# Generalizing CNNs to Graphs with Learnable Neighborhood Quantization

**Isaac Osafo Nkansah**[1]    **Neil Gallagher**[1]    **Ruchi Sandilya**[1]    **Conor Liston**[1]
**Logan Grosenick**[1]*
[1]Department of Psychiatry and BMRI, Weill Cornell Medicine,
Cornell University, New York, NY, USA

## Abstract

Convolutional neural networks (CNNs) have led to a revolution in analyzing array data. However, many important sources of data, such as biological and social networks, are naturally structured as graphs rather than arrays, making the design of graph neural network (GNN) architectures that retain the strengths of CNNs an active and exciting area of research. Here, we introduce Quantized Graph Convolution Networks (QGCNs), the first framework for GNNs that formally and directly extends CNNs to graphs. QGCNs do this by decomposing the convolution operation into non-overlapping sub-kernels, allowing them to fit graph data while reducing to a 2D CNN layer on array data. We generalize this approach to graphs of arbitrary size and dimension by approaching sub-kernel assignment as a learnable multinomial assignment problem. Integrating this approach into a residual network architecture, we demonstrate performance that matches or exceeds other state-of-the-art GNNs on benchmark graph datasets and for predicting properties of nonlinear dynamics on a new finite element graph dataset. In summary, QGCNs are a novel GNN framework that generalizes CNNs and their strengths to graph data, allowing for more accurate and expressive models.

## 1   Introduction

Many important real-world scenarios involve data structured as graphs. For example, neural networks (both biological and artificial) are typically represented as directed graphs where individual neurons propagate information to other neurons along edges. Digital networks (e.g. social networks, the internet) are graphs made up of links between digital objects, and chemical structures can be modeled as graphs made up of bonds between atoms. It can be challenging to accurately model such graph data, creating a barrier to studying these problems. Indeed, this work was motivated by our own experiments in modeling brain networks from neural data, where we have found that existing methods are often not expressive enough to effectively capture the types of phenomena we are interested in.

In recent years, the prevailing approach for learning from graph data has shifted towards methods inspired by convolutional neural networks (CNNs) [20, 21, 8, 17, 25, 4, 5]. The focus on extending the convolutional layer of CNNs to graph data is motivated by the strong and successful inductive bias of CNNs, which use trainable filters that efficiently model local structure in array-structured data (e.g., images). CNNs have been effectively employed in numerous domains, including natural language processing [26, 43] and image recognition [7]. Because graph data often exhibit strong patterns of local correlation like those seen in language and image data, it is reasonable to expect they might similarly benefit from shared local filters.

---

*To whom correspondence should be addressed: log4002@med.cornell.edu

38th Conference on Neural Information Processing Systems (NeurIPS 2024).

Early work extending CNNs to graphs focused on spectral methods [6, 3], which can suffer from high runtime and memory complexity [40]. In contrast, spatial methods aim to generalize the convolution operation explicitly from array data to graph data. Recent spatial GNN methods approach this problem by either adapting the convolution operation to graphs [11, 1, 27, 28] or by adapting graphs to fit the CNN convolutional operation [10]. Existing spatial methods, however, do not truly generalize the CNN convolution layer.

Spatial Graph Convolutional Networks (SGCNs) [5], for example, claim to generalize CNNs to graph data as during inference on array data an SGCN will have spatial filters that resemble those of a CNN. But an SGCN will not reduce to an equivalent CNN when trained on array data, and is thus not a proper generalization. This relates to a central challenge of extending convolutions to graph-structured data: in CNNs, local neighborhoods have fixed sizes and fixed ordering of the nodes within the neighborhood, a convenience that does not hold true for more general graphs. We believe that bridging the gap between CNNs and GCNs and properly generalizing the powerful local inductive bias of CNNs to GNNs will lead to improved learning for many types of graph data.

To this end, we introduce the Quantized Graph Convolution Layer (QGCL), which adds to the spatial graph neural network literature a proper generalization of the CNN convolution layer to graphs. We do this by first "quantizing" the convolution operation for CNNs into an equivalent set of non-overlapping sub-kernels applied to local geometry. Second, we describe a specific set of sub-kernels for graphs that are equivalent to a 2D convolutional kernel based on a *satisficing mapping*, which relies on relative angular displacements of nodes to quantize graph neighborhoods into sub-kernels. To generalize the QGCL to arbitrary graphs, we extend it to be able to learn neighborhood quantizations from data, using a network we term QuantNet. Furthermore, we design a residual network around the QGCL architecture, which we term Quantized Graph Residual Layer (QGRL), to make the layer more robust to model depth effects like vanishing gradients. As we were initially inspired by the regularity of widely-used finite element method (FEM) graphs, we provide a new benchmark data set for FEM (based on Navier-Stokes fluid flow on an adaptive mesh graph) and demonstrate that a QGRL-based architecture (called Quantized Graph Residual Network or QGRN) is highly competitive on such data. Next, we show that QGRNs enable competitive performance across nineteen inductive learning graph datasets. Finally, we demonstrate that incorporating QGRLs leads to superior performance in a supervised autoencoder model applied to a public EEG recordings and emotional states dataset [18].

In summary, **our main contributions** are:

1. Introducing the Quantized Graph Convolutional Network (QGCN) framework, which generalizes CNNs to graphs.
2. Empirical and formal validation that a QGCN using the *satisficing mapping* sub-kernels reduces to a 2D CNN on image graphs.
3. An end-to-end learnable quantization network (QuantNet) that extends QGCNs to arbitrary graphs.
4. A residual network inspired architecture, Quantized Graph Residual Networks (QGRNs), that further improves QGCN performance.
5. Benchmarking of QGRNs on a new Navier-Stokes FEM dataset and 19 other public benchmark graph datasets for graph classification and node classification.
6. Showing QGRLs improve joint modeling of emotional states and EEG data in a supervised autoencoder architecture.

## 2   Relevant work

**GCNs** Although inspired by spectral theory, Graph Convolutional Networks (GCNs) [17] are practically understood as a spatial GNN method as they aggregate node features within local neighborhoods (normalizing by the node degree of the central/target node) and then transform the resulting aggregated features into new features for the central/target nodes. Because all neighboring node features are scaled by fixed weights and then aggregated in the input features space, this method's number of trainable free parameters differs from CNNs (failing to generalize the CNN convolution layer). In contrast, CNNs learn embeddings of each node feature separately and the ability of each node's features to embed in a different point in the output feature space independently makes CNNs more flexible than GCNs.

**SGCNs** Spatial Graph Convolutional Networks (SGCNs) [5], a recent novel CNN-inspired GCN architecture, improve on GCNs by using graph node positional descriptors to rank nodes within their neighborhoods. They extend GCNs by using MLPs that project the relative spatial/positional descriptors of nodes into output feature space. Though the authors claim that SGCNs are equivalent to CNNs for inference, SGCNs and CNNs exhibit a different inductive bias during training. There is additional difficulty in matching CNN and SGCN model parameters and determining how scaling different parts of SGCN architecture translates into equivalent CNN adaptions. A strength of SGCNs is that they can consume pseudo-positional descriptors, making them more general than GCNs.

**KerGNNs** Kernel GNNs [9] define kernels as sub-graphs with trainable adjacency matrices and node features. The trainable node features for sub-graphs parallel CNN kernel weights, and the learned adjacency matrices allow for different topologies of sub-graph kernels to be learned. In CNNs, the adjacency matrix of the convolving kernel is fixed, hence kerGNNs generalize the CNN convolution operation well in this sense. However, the size of the direct product graph (which captures the relationship between local sub-graph patches and the convolving sub-graph kernel) grows quadratically with the sizes of the local neighborhood and the convolving sub-graph kernel. For graphs with large local neighborhoods, the computation of the adjacency matrix of the direct product graph per local neighborhood (effectively a cubic runtime complexity across the data) becomes very expensive. Further the authors suggest the use of additional trainable weights for the base random walk kernels, causing further divergence with regular CNN convolution layer.

**LGCLs** Inspired by [27], the Learnable Graph Convolutional Layer [10] approach is unlike the aforementioned methods, instead adapting graph data into a form that a regular CNN convolution operator can use. It does this by applying max pooling on the feature vectors of the local graph neighborhoods. This does not generalize the CNN convolution operation (which uses all the features within the local neighborhood and not a sub-sampled set). Further, max pooling sub-sampling constrains the model to ascribe more importance to large features; a constraint absent in CNNs.

**GATs** GATs extend the power of transformers and attention networks to graphs and have been shown to be highly performant on graph data [38, 39, 2, 36]. In current GATs, every node attends to its neighbors either in a static or dynamic fashion. The attention mechanism effectively yields edge-aware feature scalars from source nodes whose messages must be aggregated for the target node. This is akin to CNNs applying different kernel weights to different node features in local image graph neighborhoods.

**DeeperGCNs** Early works [22] adapted residual connections, inspired by *ResNets* [12], to deep graph networks to deal with the problem of vanishing gradients and over-smoothing [24]. A more recent innovation in this space, GENConv [23], builds on these residual connections (first demonstrating how powerful these connections alone are for deep networks) and innovates generalized messaging passing aggregators, learnable message normalization layers etc. to compete with state-of-the-art performance on standard graph dataset benchmarks. Other works such as DropEdge [31], which propose randomly removing graph edges, and PairNorm [44] which develops a normalization layer to tackle the problems aforementioned, are also noteworthy.

## 3 Proposed methodology

### 3.1 Extending CNNs to graphs

Quantized Graph Convolution Networks (QGCNs) are an extension of CNNs to graph data. We begin with a formal description of the convolutional layer, which is the core component of CNNs, in order to motivate the Quantized Graph Convolution Layer (QGCL). For simplicity, we focus on the convolutional layer in two dimensions with a stride size of one operating on $\mathbf{G} \in \mathbb{R}^{C' \times D' \times F'}$, where $\mathbf{G}$ is structured in a way such that proximity and adjacency in the space composed of the first two dimensions has meaning, but that the ordering of those 2D planes along the third dimension is arbitrary. As an example, 2D image data with multiple color channels exhibits this structure. In this case, a convolutional layer will generate an output feature map $\mathbf{O} \in \mathbb{R}^{C \times D \times F}$:

$$\mathbf{O}_{c,d,:} = \left( \sum_{j=0}^{J-1} \sum_{k=0}^{K-1} \mathbf{W}_{j,k,:,:} \mathbf{G}_{j+c,k+d,:} \right) + \boldsymbol{b}, \tag{1}$$

where $\mathbf{W} \in \mathbb{R}^{J \times K \times F \times F'}$ and $\boldsymbol{b} \in \mathbb{R}^F$ are the weights and bias terms of the convolutional kernel, respectively. To produce a map that is the same size as the original input, zero-padding can be added along the edges of $\mathbf{G}$ before applying the convolutional layer to result in $\mathbf{O}_{c,d,:}$ being defined for $c \in [0, C'-1], d \in [0, D'-1]$.

We can refactor the kernel parameters in Eq. 1 to have a single index $h$ iterating over the 2D space traversed by $j$ and $k$ above:

$$\mathbf{O}_{c,d,:} = \sum_{h=0}^{JK-1} \left( \hat{\mathbf{W}}_{h,:,:} \sum_{j=0}^{J-1} \sum_{k=0}^{K-1} 1_{(h=jK+k)} \mathbf{G}_{j+c,k+d,:} \right) + \hat{\boldsymbol{B}}_{h,:}, \qquad \hat{\boldsymbol{B}}_{h,:} = \frac{\boldsymbol{b}}{JK} \; \forall h, \quad (2)$$

where $\hat{\mathbf{W}}_{jK+k,:,:}$ corresponds to $\mathbf{W}_{j,k,:,:}$ in Eq. 1. The indicator function $1_{(h=jK+k)}$ has the effect of creating a mask over a single element in the 2D space defined by $j$ and $k$, with a one-to-one correspondence between each element and each value of $h$. Thus the convolutional layer can be decomposed into a set of sub-kernels (i.e., weight matrices $\hat{\mathbf{W}}_{h,:,:}$ and bias vectors $\hat{\boldsymbol{B}}_{h,:}$) along with a corresponding set of masks on the input space. This formulation of the convolutional layer allows for interesting possibilities by designing a different set of mask functions; for example, one could produce a well-defined output $\mathbf{O}$ with the same dimensions as $\mathbf{G}$ without the need for zero-padding.

In the Eq.2, the masks implicitly compare the location of elements in $\mathbf{G}$ to the relative position of the current output within the larger output tensor $\mathbf{O}$. We generalize the ideas above to graphs by allowing for masks that operate on node pairs, rather than comparing pairs of elements in tensor data. When generalizing the convolution to graph input, we want the output to be a graph as well. Here, we limit ourselves to outputting graphs with identical structure to the input and only considering the local neighborhood of a node when calculating the features of the corresponding output node. We formally define output of the quantized graph convolution layer as follows:

$$\boldsymbol{o}(v) = \sum_{h=0}^{H-1} \left( \hat{\mathbf{W}}_{h,:,:} \sum_{v' \in \mathcal{N}(v)} 1_{((v,v') \in \mathbb{M}_h)} \boldsymbol{a}(v') \right) + \hat{\boldsymbol{B}}_{h,:}, \qquad (3)$$

where $\boldsymbol{o}(v) \in \mathbb{R}^F$ provides the feature vector of the output node at the same relative location as input node $v$, $\mathcal{N}(v)$ is the set of nodes in the local neighborhood around $v$, $\mathbb{M}_h$ is the set of node pairs selected by the mask corresponding to $\hat{\mathbf{W}}_{h,:,:}$, and $\boldsymbol{a}(v) \in \mathbb{R}^{F'}$ retrieves the feature vector of node $v$. In the context of QGCNs, we refer to each $\hat{\mathbf{W}}_{h,:,:}$ as a *sub-kernel*. It is the process of using binary masks of fixed cardinality to *quantize* the space of potential nodes in a local neighborhood that gives quantized graph convolution networks their name. This framework is sufficient to include most practical use cases of the convolutional layer. For example, convolutional kernels with any dimension larger than three can be represented by considering tensor elements to be connected in the graph representation if the kernel would apply to one node while the kernel is positioned with the other node at its 'center'. Two noteworthy exceptions that do not fit in the QGCN framework are stride sizes larger than one or convolutional kernels with odd dimensions (to be explored in future work).

## 3.2 A satisficing mapping generalizes local convolutional kernel masks

In this section, we show how the sub-kernel masks ($\mathbb{M}_h$) associated with convolutional kernels can be extended to the case of graphs with (pseudo-)positional information. The masks associated with CNN convolutions are functions of the relative position of tensor elements (nodes) to the position of the current element in the output tensor (see Eq. 2). We refer to these as the *natural* convolutional masks (see Fig. 1 f). For graphs with positional information, it is possible to formalize a method for choosing a set of sub-kernel masks that would produce standard convolutional layer masks when applied to tensor data that has been converted to graph form. For simplicity we consider the case of a convolution in two dimensions where data is converted to a 2D positional graph by assuming that edges exist only between adjacent elements (nodes) and extend this case to handle all 2D positional graphs. Note that in the absence of positional information, any other information associated with nodes that can be embedded into a 2D space can be treated as pseudo-positional information to enable this more general approach.

As seen in Eq. 2, each mask is defined by the indicator function $1_{(h=jK+k)}$, which selects a single element. When dealing with tensor data, the relative position of the elements selected by these masks

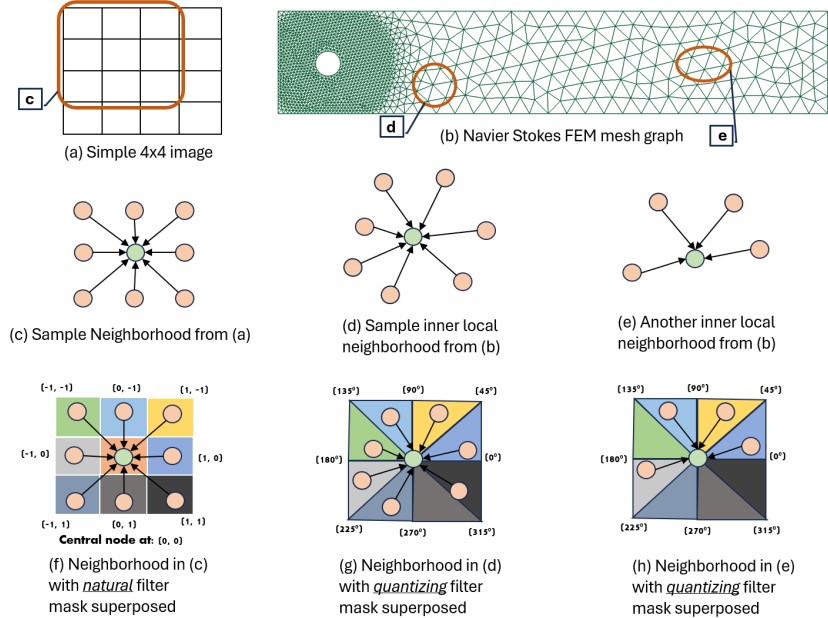

Figure 1: Contrasting the assignment of kernel weights to local neighborhood nodes for traditional CNN convolution kernels and the satisficing mapping sub-kernels of a QGCL layer. Traditional CNN convolution kernel is depicted with its natural kernel weights masks while QGCL sub-kernels are shown with their corresponding quantizing kernel masks on graph neighborhoods. Note that the angular quantization bins have inclusive angular lower bounds and exclusive angular upper bounds, such that nodes falling on the edges are mapped to unique sub-kernels (e.g., the node in (h.) on the 135°edge maps to the green mask sub-kernel.

remains fixed as the convolutional kernel moves to different output positions. In contrast, we cannot assume that the nodes in a local neighborhood will always be located in the same relative positions for all local neighborhoods within a graph. Because we desire a set of masks that is applicable to all 2D positional graphs, our masks must define a way to potentially map all of 2D space to a set of sub-kernels. One simple option is dividing 2D space into regularly spaced non-overlapping segments defined by angular position relative to the center node of the local neighborhood that the map is being applied to. Then the center-neighbor node pair $(v, v')$ is part of the $h^{th}$ sub-kernel mask ($\mathbb{M}_h$) when

$$
\mathbb{M}_h = \left\{ (v, v') \mid 2\pi \frac{h-1}{H-1} + \phi \leq \theta(v, v') < 2\pi \frac{h}{H-1} + \phi \right\}_{\forall v \in V, v' \in \mathcal{N}(v)}, \ h \in [1, H-1],
$$

$$
\theta(v, v') = \tan^{-1}\left( \frac{p_y(v') - p_y(v)}{p_x(v') - p_x(v)} \right) \in [0, 2\pi),
$$

(4)

where $\theta$ is the angle of the neighbor node relative to the center node, $p_x$ and $p_y$ return the $x$ and $y$ coordinates of a node, respectively, and $H$ is the total number of sub-kernels. The $0^{th}$ sub-kernel is applied only to the node pair $(v, v)$ made up of the center node and itself. The offset angle $\phi$ is an optional hyperparameter that can be used to select the starting point from which the space is divided. To select $H$, we choose the smallest number that results in all nodes within a local neighborhood being assigned to a different sub-kernel, which we refer to as a *satisficing mapping*. It is easy to see that separating the local elements of tensor data based on this method produces the same assignments as the natural convolutional kernel masks (Fig. 1 f). Algorithm 2 in Appendix D outlines an efficient process for determining the value of $H$ that satisfies this condition. If the sub-kernel masks are chosen in this way and, importantly, the bias values for each sub-kernel are tied to the same value (i.e. $\hat{B}_{h,:} = \frac{b}{JK}, \forall h$), then we arrive at a set of quantized graph convolution sub-kernels that will behave on 2D positional graph data identically to a standard 2D convolutional layer on equivalent tensor data (see Appendix C for proof and below for empirical validation).

With this *satisficing mapping* approach, the process of assigning nodes in every local neighborhood to sub-kernels (see Algorithm 1 in Appendix D) incurs a computational cost of $O(|V|^2)$ in each forward pass. In the case of homogeneous graph meshes, choosing to cache the satisficing mapping incurs at worst $O(|V|^2)$ space complexity. Using Algorithm 2 for determining the minimum number of subkernels for the convolution such that a satisficing mapping is honored adds a constant cost on top of Algorithm 1. We note that this is only one of many possible ways in which the natural convolutional mask and sub-kernels described in Eq. 2 can be extended to positional graph data.

### 3.3 Learning neighborhood quantization

Next, we introduce a method for generating masks that assign nodes to sub-kernels in arbitrary dimensions and regardless of whether positional information is present. Specifically, we frame quantization as a learnable multinomial classification problem where for a learnable model assigns a sub-kernel to each center-neighbor node pair in a local neighborhood. This approach was inspired by the idea of dilated convolutions in CNNs, akin to learning the spacings of the kernel elements during CNN convolution [15]. To learn the quantization, we introduce *QuantNet*, an MLP that projects node features or (pseudo-)positional descriptors into a higher dimensional space where we difference the target and source features and then project this difference to a vector representing assignment weights for each sub-kernel (see Fig. 2). The mask $\mathbb{M}_h$ associated with sub-kernel $h$ contains the ordered node pair $(v, v')$ when *QuantNet Q* assigns the node pair to $h$:

$$
\begin{aligned}
\mathbb{M}_h &= \{(v, v') | Q(v, v') = h\}_{\forall v \in V, v' \in \mathcal{N}(v)}, \\
Q(v, v') &= \mathrm{argmax}(\mathrm{softmax}(U_2(U_1(v; \eta) - U_1(v'; \eta); \theta))),
\end{aligned}
\tag{5}
$$

where $U_1$ is a high dimensional MLP projector with parameters $\eta$ for the input features (spatial descriptors, node features, etc.), $U_2$ is a low dimensional projector with parameters $\theta$ for the difference in high dimensional features projected by $U_1$, $v$ is a node in $V$ (the node set of the input graph), and $\mathcal{N}(v)$ denotes the local neighborhood node set of $v$. Note *argmax* in Eq. 5 is symbolic; it represents any differentiable function that outputs discrete categorical samples, for example, a custom *argmax* implemented with a straight-through gradient estimator or Gumbel-Softmax with hard sampling (our implementation uses Gumbel-Softmax with hard sampling) [13]. The *QuantNet* network architecture is shown in Fig. 2. Finally, we reiterate that because *QuantNet* can use any vector in place of positional information, QGCL becomes extensible to graphs without explicit positional information.

### 3.4 Integrating QGCNs with a residual architecture

A common and successful approach used to address vanishing gradients and over-smoothing in GNNs is residual learning, inspired by the success of ResNets for CNNs [12]. We adapt this framework

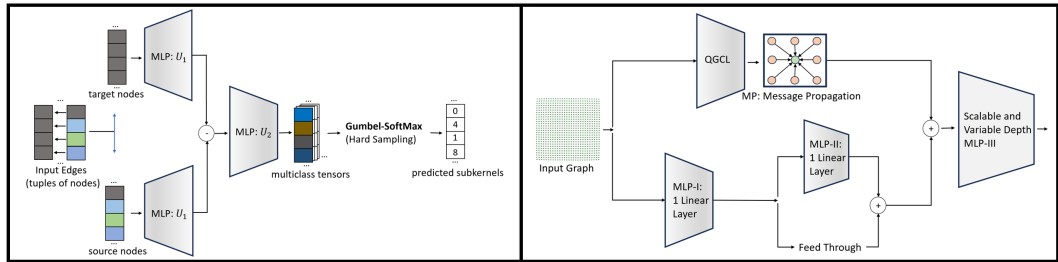

Figure 2: *QuantNet and Quantized Graph Residual Layer (QGRL).* [Left] A learnable network for dynamic quantization of nodes to subkernels in different local neighborhoods. The message passing framework in PyTorch provides the source and target nodes across all edges so QGCL doesn't have any computation overheads in defining the input tensors fed into QuantNet. The output of QuantNet is the satisficing mapping used to filter the receptive fields of the QGCL subkernels. [Right] An architectural retrofit of QGCL, incorporating 2 residual blocks: (1) outer residual block for the QGCL and (2) an inner residual block for learning features from input graph messages. The network combines all features dynamically via MLP-III to prepare the final node messages for the layer.

Table 1: *Standard image datasets*. CNN and QGCN model accuracies (mean $\pm$ S.D.).

| Dataset | Test Accuracy | |
| --- | --- | --- |
| | CNN | QGCN |
| MNIST | $98.92 \pm 0.10$ | $98.98 \pm 0.04$ |
| FashionMNIST | $92.56 \pm 0.18$ | $92.39 \pm 0.13$ |
| CIFAR-10 | $80.21 \pm 0.29$ | $79.59 \pm 0.35$ |

Table 2: *Custom Graph Datasets*. QGRN and SGCN Performance Comparison

| Dataset | Parameters (k) | | FLOPs (M) | | Test Accuracy (%) | |
| --- | --- | --- | --- | --- | --- | --- |
| | QGRN | SGCN | QGRN | SGCN | QGRN | SGCN |
| NS-Binary | 58.67 | 57.37 | 535.17 | 800.32 | $\mathbf{99.67} \pm 00.23$ | $\mathbf{99.67} \pm 00.23$ |
| NS-Denary-1 | 58.67 | 57.37 | 535.17 | 800.32 | $\mathbf{97.47} \pm 01.01$ | $94.23 \pm 01.32$ |
| NS-Denary-2 | 58.67 | 57.37 | 535.17 | 800.32 | $\mathbf{95.13} \pm 05.21$ | $93.33 \pm 05.77$ |
| AIDS | 59.43 | 57.61 | 9.30 | 13.03 | $\mathbf{99.50} \pm 00.14$ | $99.25 \pm 01.07$ |
| Letter (high) | 56.33 | 49.72 | 0.77 | 0.98 | $\mathbf{94.10} \pm 00.81$ | $93.21 \pm 00.79$ |
| Letter (low) | 53.14 | 49.72 | 0.74 | 0.95 | $\mathbf{99.81} \pm 00.23$ | $99.62 \pm 00.14$ |
| Letter (med) | 53.14 | 49.72 | 0.74 | 0.95 | $\mathbf{97.14} \pm 00.71$ | $95.24 \pm 00.79$ |

to QGCNs, arriving at the architecture shown in the right panel of Fig. 2, which we call *Quantized Graph Residual Layer (QGRL)*. Notice that QGRL subsumes and generalizes QGCL.

## 4 Experiments

### 4.1 Empirical validation of equivalence with CNNs on 2D images

First, we confirmed that the QGCL performs similarly to the CNN convolutional layer when applied to image data. We considered three standard 2D image datasets that vary in complexity: MNIST [21], Fashion-MNIST [41], and CIFAR10 [19]. MNIST contains gray-scaled images of handwritten digits of shape 28x28x1, FashionMNIST contains fashion images of shape 28x28x1, and CIFAR10 consists of color images of shape 32x32x3 from 10 categories. Figure 4 in Appendix E shows the different CNN models trained for the different standardized datasets. We created a 3-layer CNN and its equivalent QGCN model for the MNIST dataset, 6-layer network models for Fashion-MNIST, and 9-layer network models for CIFAR10. The equivalent QGCN models have the same architecture as the CNN models, except that QGCN uses QGCL layers internally in place of traditional convolutional layers. All models were trained 5 times on each dataset, with different random parameter initializations and random ordering of the training data for each run, using cross-entropy loss and the Adam optimizer [16] with a learning rate of 0.01 for 200 epochs. In order to establish equivalence between CNN and QGCN while avoiding full-dataset ceiling effects we separately trained models fit at three different sample sizes (yielding different bias-variance trade-offs) by varying the dataset train-test splits (see Appendix F).

Table 1 shows how QGCN performs almost identically to CNN across the different standard image datasets. Appendix F (table 8) shows the expanded version of 1, showing different train-test splits, devised to explore bias-variance trade-offs. Additionally, Appendix G shows training loss and train/test set accuracy profiles over a wide range of learning rates for both CNN and its parameter-matched equivalent QGCN (not QGRN) to show how model behaviors are very similar even in different bias-variance trade-off regimes. These results confirm how both models follow exceedingly similar loss trajectories during training and have the same accuracy profiles, empirically supporting our formal proof of CNN and QGCN equivalence on image data.

### 4.2 Graph Classification: Datasets with Positional Descriptors

Next, we compared QGRN to SGCN on graph datasets that have positional descriptors, including a novel FEM fixed-mesh graph dataset. The graph benchmark datasets: AIDS, Letters (high/low/med) were post-processed to extract out their positional node descriptors into separate positional attributes

Table 3: *Graph kernels benchmark datasets - I.* Test Accuracy (%) across different GCNs

| Models | AIDS | Frankenstein | Mutag | Proteins |
|---|---|---|---|---|
| QGRN | **99.50** $\pm$ 0.10 | **75.58** $\pm$ 0.40 | **99.99** $\pm$ 00.26 | **80.20** $\pm$ 00.14 |
| GCNConv | 90.92 $\pm$ 0.38 | 60.27 $\pm$ 0.06 | 92.68 $\pm$ 02.29 | 71.95 $\pm$ 00.57 |
| ChebConv | 93.42 $\pm$ 0.14 | 62.56 $\pm$ 0.28 | 91.87 $\pm$ 01.41 | 75.58 $\pm$ 01.51 |
| GraphConv | 94.25 $\pm$ 0.16 | 65.89 $\pm$ 0.28 | 95.12 $\pm$ 00.48 | 74.59 $\pm$ 00.57 |
| SGConv | 91.92 $\pm$ 0.14 | 60.23 $\pm$ 0.06 | 92.68 $\pm$ 00.49 | 72.94 $\pm$ 00.57 |
| GENConv | 99.17 $\pm$ 0.14 | 66.74 $\pm$ 0.42 | 98.37 $\pm$ 01.41 | 79.87 $\pm$ 00.57 |
| GeneralConv | 94.33 $\pm$ 0.14 | 65.67 $\pm$ 0.42 | 92.68 $\pm$ 00.45 | 74.59 $\pm$ 00.57 |
| GATv2Conv | 98.58 $\pm$ 0.38 | 63.71 $\pm$ 0.29 | 95.94 $\pm$ 01.41 | **80.20** $\pm$ 00.99 |
| TransformerConv | 99.25 $\pm$ 0.14 | 64.40 $\pm$ 0.32 | 92.68 $\pm$ 02.29 | 79.21 $\pm$ 00.67 |

that QGRN and SGCN use. This was to show that QGRN is able to use positional descriptors when they exist and is able to perform competitively with models such as SGCN designed specifically to use positional descriptors. Table 2 provides test set accuracy, as well as model size and computational complexity for each model on each positional graph dataset. We highlight that QGRN performs equal to or better than SGCN on all positional graph datasets we tested.

### 4.2.1 Custom FEM Dataset

We compare QGRNs and matched SGCNs on our new simulated Navier-Stokes non-linear dynamics benchmark datasets for binary and denary classification. We simulated the "flow past a cylinder" problem on an adaptive mesh with the underlying two-dimensional flow geometry depicted in Appendix H Fig. 15a. For binary classification, we separated laminar and turbulent flows based on distinct Reynold's number ($Re$) values while for denary classification we used evenly spaced $Re$ values. We created three datasets: Navier-Stokes-Binary (NS-Binary) for easier binary classification and Navier-Stokes-Denary-1 (NS-Denary-1) and Navier-Stokes-Denary-2 (NS-Denary-2) for more challenging denary classification, with NS-Denary-2 being most challenging (most closely spaced $Re$ values; see Appendix H). QGRNs matched SGCN performance on the binary task and outperformed SGCNs on the more challenging denary tasks.

### 4.3 Graph Classification: Generic Graph Datasets

Finally, we compared QGRNs to matched (in model parameter count) GNN models using inductive learning datasets from Benchmark Data Sets for Graph Kernels [14], namely: AIDS, COIL-DEL, Frankenstein, Enzymes, Letter (low/med/high), Mutagenicity, Proteins, Proteins-Full, Mutag and Synthie. See Appendix I for a description of each dataset. We trained on a number of novel GNN architectures including Transformer networks (GAT, TransformerConv), showing how QGRN maintains superior performance over its competitors on many of the benchmark datasets (which lack positional descriptors). We size all models relative to QGRN to have a matched number of parameters for fair comparison. Given the simplicity of some of the models, this effort of establishing equivalence yields slightly different architectures, however, all architectures are constrained to have the same depth. More details are provided in the dataset configuration section of the provided code. All models were trained with the Adam optimizer using cross-entropy loss for 500 epochs at 4 different learning rates (0.1, 0.01, 0.001, 0.0001). Appendix subsection K shows how QGCN wall clock time varies compared to other GNN methods with matching parameter sizes. Each run is repeated 3 times and we report the best accuracy for each model across these learning rates.

Tables 3, 4 and 12 showcase QGRN matching and outperforming all GNN methods across a diverse sampling of inductive graph learning problems. All datasets appearing in these tables either do not have positional descriptors or have their positional attributes collapsed into the individual node features. We do this because many of the generalized GNNs in the literature such as ChebConv, GCNConv, GraphConv etc. are not able to handle positional descriptors as separate attributes from node features. In the tables, we see clearly how QGRN matches or outperforms all models on all benchmark graph classification tasks.

Finally, there are additional experiments we carried out such as how QGRN fares in deeper networks (see Appendix O), how different quantizations impact model performance (see Appendix P), how

Table 4: *Graph kernels benchmark datasets - II*. Test Accuracy (%) across different GCNs

| Models | Synthie | Letters (high) | Enzymes | Coil-Del |
|---|---|---|---|---|
| QGRN | **99.99** $\pm$ 0.23 | **94.10** $\pm$ 0.81 | **72.50** $\pm$ 00.96 | **94.14** $\pm$ 00.78 |
| GCNConv | 80.89 $\pm$ 0.70 | 40.76 $\pm$ 0.29 | 43.61 $\pm$ 00.48 | 20.42 $\pm$ 00.39 |
| ChebConv | 87.40 $\pm$ 0.70 | 55.56 $\pm$ 0.40 | 48.06 $\pm$ 01.74 | 27.86 $\pm$ 00.29 |
| GraphConv | 89.43 $\pm$ 0.70 | 54.54 $\pm$ 0.22 | 46.94 $\pm$ 00.48 | 26.91 $\pm$ 00.30 |
| SGConv | 80.08 $\pm$ 0.41 | 40.44 $\pm$ 0.11 | 44.44 $\pm$ 01.27 | 20.81 $\pm$ 00.44 |
| GENConv | 92.28 $\pm$ 0.86 | 91.62 $\pm$ 0.83 | 67.22 $\pm$ 02.93 | 81.91 $\pm$ 00.86 |
| GeneralConv | 91.46 $\pm$ 0.22 | 54.54 $\pm$ 0.11 | 48.61 $\pm$ 00.48 | 28.33 $\pm$ 00.60 |
| GATv2Conv | 92.68 $\pm$ 0.11 | 76.83 $\pm$ 0.23 | 53.89 $\pm$ 03.16 | 65.48 $\pm$ 01.82 |
| TransformerConv | 92.68 $\pm$ 0.11 | 85.52 $\pm$ 0.29 | 54.17 $\pm$ 01.34 | 62.71 $\pm$ 00.76 |

Table 5: *Node Classification Datasets*. Test Accuracy (%) across different GNNs

| Models | Homophilic | | | Heterophilic | |
|---|---|---|---|---|---|
| | Computers | Cora | PubMed | Chameleon | Squirrel |
| QGRN | 90.02 $\pm$ 0.02 | **89.02** $\pm$ 0.14 | **89.11** $\pm$ 0.15 | 74.15 $\pm$ 0.37 | 56.17 $\pm$ 0.45 |
| GraphConv | 87.96 $\pm$ 0.16 | 87.15 $\pm$ 0.44 | 88.39 $\pm$ 0.07 | 72.77 $\pm$ 0.32 | 64.25 $\pm$ 0.09 |
| GENConv | **91.66** $\pm$ 0.05 | 86.31 $\pm$ 0.36 | 87.73 $\pm$ 0.19 | 71.56 $\pm$ 0.67 | 58.00 $\pm$ 0.18 |
| GeneralConv | 89.29 $\pm$ 0.02 | 87.64 $\pm$ 0.04 | 88.97 $\pm$ 0.09 | **78.11** $\pm$ 0.29 | **66.80** $\pm$ 0.08 |
| EGConv | 91.50 $\pm$ 0.06 | 88.34 $\pm$ 0.30 | 88.38 $\pm$ 0.08 | 63.54 $\pm$ 0.07 | 48.44 $\pm$ 0.41 |

to determine the number of quantization bins for a given dataset (see Appendix Q) and finally a comparison of QGRN with leaderboard performances from papers with code (see Appendix M).

## 4.4 Node Classification

Next, we evaluated the performance of QGRNs on various node classification tasks. We tested on multiple types of node classification datasets, including citation networks (like Cora, PubMed), Wikipedia hyperlinks networks (like such as the Chameleon dataset) and product relations networks (such as Amazon Computers) [33, 35, 42]. We highlight that the datasets used in this exploration exhibit different degrees of homophily and heterophily properties. The Chameleon and Squirrel datasets exhibit strong heterophily while all the others exhibit stronger homophily. We chose a single architecture that proved reasonably performant across all the models we compared against (see Figure 5 in Appendix E). We include another novel GNN, EGConv [37] to highlight the competitiveness of our method to recent methods. We trained all models across a range of learning rates (0.1, 0.05, 0.01, 0.005, 0.001) for 2000 epochs and mimicked early stopping by caching the model state that produced the largest validation set accuracy. Given the now apparent fact that Message Passing Neural Networks generally struggle with heterophilic datasets, we designed the generic architecture with edge directionality awareness, as inspired by authors Rossi et. al. [32].

We see in Table 5 and Tables 18 and 19 in Appendix L, that QGRNs performed competitively across the homophilic datasets and appreciably well on the two heterophilic datasets. On the Chameleon heterophilic dataset QGRN performs moderately well, outperforming all comparison models except GeneralConv. The Squirrel dataset, which is the most heterophilic of those we tested, proved more challenging for our QGRN models. Overall, QGRNs perform reasonably well on node classification tasks, especially in cases where the dataset exhibits homophilic properties.

## 4.5 Supervised Autoencoder Model of Emotional States in EEG data

A major motivation behind these models was the need for more expressive ways to model graphical data with positional information captured from complex geometrical surfaces. As a demonstration of the practical value of QGRNs beyond standardized benchmark datasets, we developed a supervised autoencoder (SAE) to model brain networks related to valence in the publicly available DEAP dataset [18]. The last 42 s of each recording in the DEAP EEG dataset were divided into sliding windows of 3 s with 50% overlap, then were z-scored and spectral power was calculated in four

Table 6: *EEG SAE test set performance*. All values are presented as the mean±SEM over all subjects.

| Models | QGRN | SGCN |
|---|---|---|
| MSE loss (Generative) | **1787.33** $\pm$ 315.51 | $2169.38 \pm 317.80$ |
| CE loss (Supervised) | **0.65** $\pm$ 0.01 | $0.66 \pm 0.01$ |
| AUC | **0.59** $\pm$ 0.01 | $0.56 \pm 0.01$ |

frequency bands (4-8 Hz, 8-12 Hz, 12-30 Hz, and 30-44 Hz) for each of the 32 electrodes. Power features were used as node attributes in a fully-connected graph containing all electrodes. We trained a separate model for each subject, dividing the data for each subject according to a 64%/16%/20% train/validation/test split. The generative objective for the autoencoder was mean-squared error (MSE) and the supervised objective was cross-entropy (CE) for classifying whether subjects self-rated their emotional state as positive valence ($\geq 5$) or negative valence ($< 5$) on a 9-point scale. Models were pre-trained for 1000 epochs on just the generative objective, then for another 100 epochs with one of 3 values of weight (100, 1000, or 10000) on the classification objective. Validation sets were used to select the weight and number of training epochs with highest area under the receiver operating curve (AUC). As shown in table 6, we found that using QGRN layers in the same autoencoder architecture compared to SGCN layers resulted in better generative and supervised loss values on the held out test sets. In addition, the QGRN-based model resulted in appreciably better classification performance (measured by AUC) for this difficult classification problem.

## 5 Discussion

This work introduces Quantized Graph Convolution Networks (QGCNs), a flexible framework for designing graph neural networks that extends the benefits of CNNs' strong local inductive bias to graphs. QGCNs "quantize" the space of possible neighbor nodes in a local neighborhood into a fixed set of sub-kernels. We show, both theoretically and empirically, that QGCNs are a generalization of CNNs to graphs with positional information. We extend QGCNs to arbitrary graphs by introducing the QuantNet method for learning sub-kernel assignment in a QGCN. We then show that embedding a QGCL within a residual network architecture gives state-of-the-art results on a suite of benchmark graph and node classification tasks, in addition to a novel Navier-Stokes FEM dynamics classification dataset. Finally, we demonstrate that QGCLs improve the performance of a supervised autoencoder to jointly model EEG data and emotional state.

One significant limitation of the current work is that the implementation of the QGCL is not yet efficient as demonstrated in the comparison of wall clock runtime with other models in Appendix K. Future work will look into parallelized subkernel operations to demonstrate wall clock runtime competitive with similar models in the literature. QGCNs also do not generalize *all* CNNs, as it cannot represent convolutional layers with odd-numbered kernel sizes (i.e. no 'center' element) or stride sizes other than one. Finally, the use of the QGCL layer in more complex architectures such as U-Nets could be explored in future work, alongside examining the performance of this model on more inductive and transductive tasks to assess its strength in various learning scenarios.

We have shown that QGCNs are a true generalization of CNNs and therefore are capable of maintaining the same powerful inductive bias that has led to such great success in the application of CNNs. In our experience, QGCNs are more expressive and efficient at learning complex local patterns of correlation in graph data than competing methods, and we expect that QGCNs can be successfully applied in many domains where graph data are prevalent. We expect that future research into different masking functions for QGCN sub-kernels (in addition to the angular satisficing mapping and QuantNet masking functions we outline here) will further extend the potential usefulness of QGCNs.

## Acknowledgments and Disclosure of Funding

NG is supported by the NIH CTSA TL1 training program via NIH/NCATS Grant #2TL1TR2386. LG is supported by NIH R01MH131534, R01MH118388, New Venture Fund 202423, a Whitehall Foundation grant (WF 2021-08-089), a Cornell Center for Pandemic Prevention Research seed grant, and an A2 Collective pilot grant (PennAITech, NIA).

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

## A    Code

Please find the code-base for the paper here: `https://github.com/Grosenick-Lab-Cornell/QuantNets`

## B    Nomenclature disambiguation

The table 7 is meant to assist with disambiguating different layer, network and model names introduced in this paper.

Table 7: *Glossary*. Terms used in this paper and quick reference meaning

| **Name** | Expanded | Comment |
|---|---|---|
| QGCL | Quantized Graph Convolution Layer | CNN-equivalent convolutional layer that can be applied to graphs with (pseudo-)positional information |
| QGCN | Quantized Graph Convolution Network | A network / model that subsumes the QGCLs |
| QuantNet | Quantizing Network | A learnable network for assigning nodes in local neighborhoods to QGCL sub-kernels |
| QGRL | Quantized Graph Residual Layer | QGCL retrofitted with residual network architecture |
| QGRN | Quantized Graph Residual Network | A network / model that subsumes QGRLs |

## C    The *satisficing mapping* QGCL for 2D positional graphs extends the 2D local convolutional layer

In this section we formally prove that the *satisficing mapping* QGCL described in Section 3.2 is identical to the 2D convolutional kernel on the local neighborhood of array elements (i.e. $3 \times 3$ kernel) as described in Section 3.1.

**Theorem 1.** *The satisficing mapping QGCL is identical to a $3 \times 3$ convolutional layer when applied to 2D array data converted to a 2D positional graph by forming edges between adjacent elements (i.e. nodes).*

*Proof.* We have already defined the output of the *satisficing mapping* QGCL in (3) and (3.2); and we have shown that the output of a standard convolutional layer is given by (2) in Section 3.1. Here, it is sufficient to show that these two are equivalent when using 2D array data that is represented as a 2D positional graph.

$$\boldsymbol{o}(v) = \sum_{h=0}^{8} \left( \hat{\mathbf{W}}_{h,:,:} \sum_{v' \in \mathcal{N}(v)} 1_{((v,v') \in \mathbb{M}_h)} \boldsymbol{a}(v') \right) + \hat{\boldsymbol{B}}_{h,:} \tag{6}$$

$$= \sum_{h=0}^{8} \left( \hat{\mathbf{W}}_{h,:,:} \sum_{j=0}^{2} \sum_{k=0}^{2} 1_{(h=3j+k)} \boldsymbol{a}(v'_{j+c,k+d}) \right) + \hat{\boldsymbol{B}}_{h,:} \tag{7}$$

$$= \sum_{h=0}^{8} \left( \hat{\mathbf{W}}_{h,:,:} \sum_{j=0}^{2} \sum_{k=0}^{2} 1_{(h=3j+k)} \mathbf{G}_{j+c,k+d,:} \right) + \hat{\boldsymbol{B}}_{h,:} \tag{8}$$

$$= \sum_{h=0}^{8} \left( \hat{\mathbf{W}}_{h,:,:} \sum_{j=0}^{2} \sum_{k=0}^{2} 1_{(h=3j+k)} \mathbf{G}_{j+c,k+d,:} \right) + \frac{\boldsymbol{b}}{9} \tag{9}$$

$$= \mathbf{O}_{c,d,:} \tag{10}$$

The proof begins by restating the definition of the QGCL in (6). As described in Section 3.2, the sub-kernel masks are chosen such that within each local neighborhood $\mathcal{N}(v)$ each mask $\mathbb{M}_h$ is selective of a single neighbor node $v'$. Thus we can iterate over each sub-kernel and mask by iterating over the rows and columns of the equivalent convolutional kernel, giving (7). We use $v'_{a,b}$ to represent the node associated with the array data $\mathbf{G}_{a,b,:}$, which gives (8). In Section 3.2, we define the bias vectors associated with each sub-kernel $\hat{\boldsymbol{B}}_{h,:} = \frac{\boldsymbol{b}}{JK}, \forall h$, giving (9). This is equivalent to the form given in (2) in Section 3.1, completing our proof. $\qquad\square$

This equivalence indicates that both types of convolutional models will produce identical output vectors during inference when given the same parameters ($\hat{\mathbf{W}}$, $\boldsymbol{b}$) and input data ($\mathbf{G}$). This also means that gradients with respect to the parameters will be identical for the purposes of backpropagation, because $f(x) = g(x)$ implies $\frac{\partial f}{\partial x} = \frac{\partial g}{\partial x}$.

## D  Quantizing local neighborhoods with satisficing mapping

---
**Algorithm 1** Quantization algorithm: pseudo-code

---
**Input:** $G = (V, E, P)$ and $S$, where $V$ is the
node set of $G$, $E$ is the adjacency matrix of $G$,
$P$ is the positional descriptors for all nodes
in $V$ and $S$ is the list of subkernels
and their angular centroids or quantization ranges.
**Helper Functions:**
$exnb$ - extracts local graph neighborhoods
$crad$ - computes relative angular distances
$gski$ - gets subkernel (by index) a node maps to
$M \leftarrow \{\}$
**for** $a \in V$ **do**
$\quad N_a \leftarrow \text{exnb}(G, a)$
$\quad$ **for** $b \in N_a$ **do**
$\quad\quad r \leftarrow \text{crad}(a, b)$
$\quad\quad k\_index \leftarrow \text{gski}(S, r)$
$\quad\quad M.\text{add}( \{b, k\} )$
**return** $M$

---

Algorithm 1 shows in pseudo-code the operation to be done deterministically in each forward pass iteration of the QGCL model with *satisficing mapping kernels*. Notice that we have $O(k|V|^2)$, where $|V|$ is the cardinality of the node set of the graph and $k$ is the constant overhead of computation in each neighborhood to map nodes to specific kernels: this is the complexity of the function named $gski$. For homogeneous graphs with fixed graph meshes, this computation could be cached, adding a memory/space cost of $O(|V|^2)$, exemplified in the strongly connected graph scenario.

### D.1 Determining the minimum number of sub-kernels per QGCL layer

---

**Algorithm 2** QGCL minimum sub-kernel number

---

**Input:** $G = (V, E, P)$ and $ub$, where $V$ is the node set of $G$, $E$ is the adjacency matrix of $G$, $P$ is the positional descriptors for all nodes in $V$ and $ub$ is the upper-bound on number of sub-kernels in the QGCL layer.

**Helper Functions:**
$mnd$ - extracts graph's max node degree
$exnb$ - extracts local graph neighborhoods
$absk$ - assigns angular bins to sub-kernels
$crad$ - computes relative angular distances
$ansk$ - assigns neighborhood nodes to sub-kernels
$inai$ - determines if neighborhood assignment is injective
$M \leftarrow \text{mnd}(G)$
**if** $M \geq ub$ **then**
    **return** $ub$
**else**
    $N \leftarrow \text{exnb}(E)$
    **for** $i = M$ until $ub$ **do**
        $k \leftarrow i$                                             ▷ k - number of sub-kernels
        $q \leftarrow \frac{360}{k}$
        $b \leftarrow \text{absk}(k, q)$
        **for** $n \in N$ **do**
            $a \leftarrow \text{crad}(n)$
            $is\_injective \leftarrow \text{inai}(\text{ansk}(a, b))$
            **if** not $is\_injective$ **then**
                **break**
        **if** $is\_injective$ **then**
            **return** $k$
    **return** $ub$

---

We introduce Algorithm 2 to establish the minimum number of sub-kernels necessary to ensure satisficing mapping across any arbitrary static graph dataset. This algorithm runs once during the first training epoch to initialize the minimum number of sub-kernels necessary for the particular dataset. This cost occurs just once for the very first batch of training data; afterwards the mapping of sub-kernels to local neighborhood nodes can be cached so that subsequent batches and nodes can reuse it. The runtime complexity of the algorithm is $O(|V|^2)$, where $|V|$ represents the cardinality of the graph node set. This worst-case asymptotic complexity is observed in strongly connected graphs. The quadratic dependency on $|V|$ arises from the inner for loop, which iterates through each node within the graph (resulting in an $O(|V|)$ operation) and computes relative angular displacements for each local neighborhood (another $O(|V|)$ worst-case-complexity operation in the case of strongly connected graphs) to assign nodes to their respective sub-kernels. Caching incurs $O(|V|^2)$ space cost in the case of strongly connected graphs where the same node in every neighborhood maps to a different subkernel.

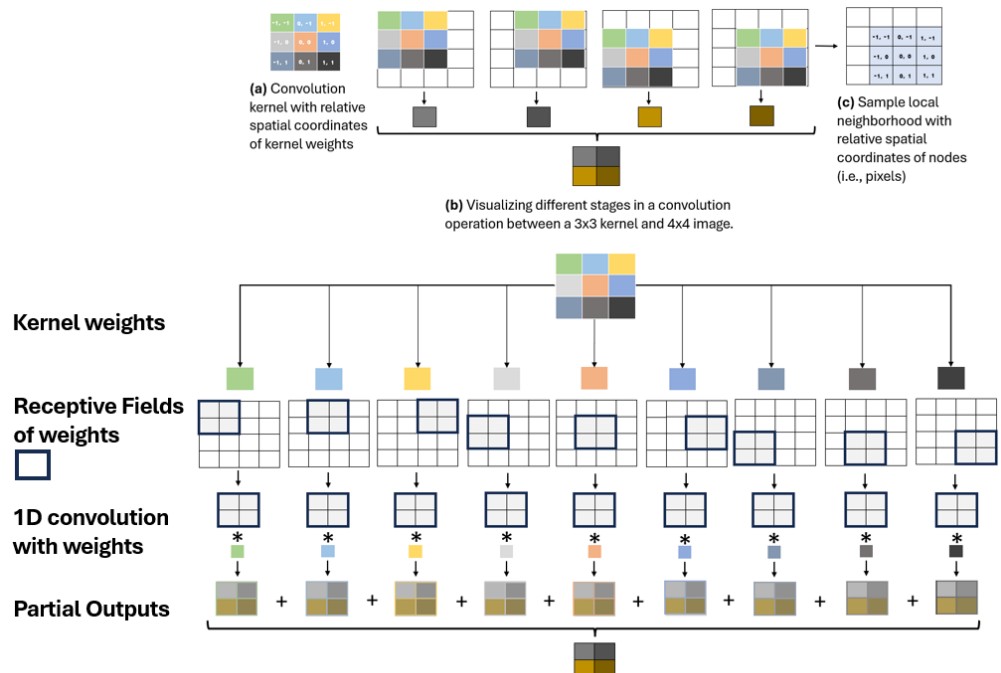

Figure 3: Visualization of natural ranking of nodes within local neighborhoods of image graph data via relative positional descriptors, which imposes a natural relative positional descriptor label on the convolving sub-kernels.

# E   Model Architectures

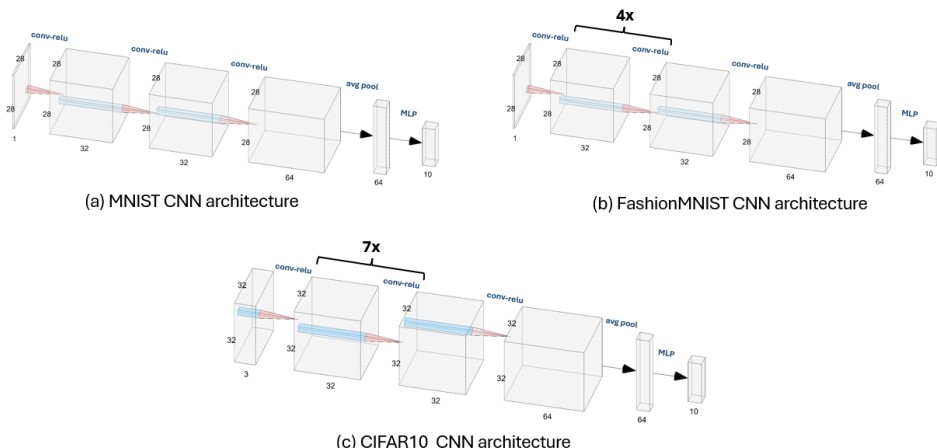

Figure 4: CNN models trained for different standardized datasets: MNIST, FashionMNIST and CIFAR10. QGCN and SGCN architectures were exactly the same as the CNN architectures except with the convolutional layers replaced with QGCL and SGCN layers.

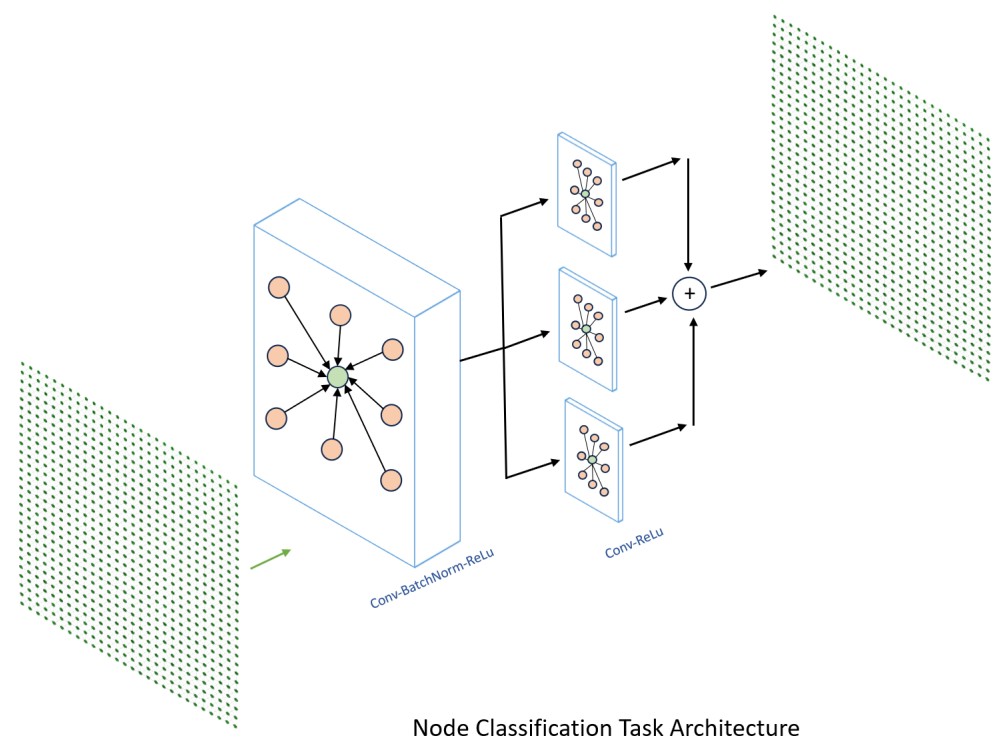

Node Classification Task Architecture

Figure 5: Neural GNN Architecture for Node Classification Task. All GNNs we tested had the same architecture as depicted. The generic architecture has 2 blocks, the first is a convolutional layer followed by batch normalization and then a ReLU activation function. The second block in the architecture sums up features from three identical blocks, each of which is a convolutional layer followed by a ReLU activation function. For any given GNN, it's message passing layer was substituted into the convolutional layer, depicted in the figure, to derive the overarching architecture. In doing so, we guaranteed an iso-architecture comparison.

# F  Standard Datasets: Full Results

We trained CNN and QGCN models on sub-sampled splits of the standard image datasets described in the main paper. The splits chosen were: (train, test) = (100, 20), (1000, 200) and (10000, 1000), with equal sampling across categories. These sub-sampled versions of the data allowed a greater variance in model performance, facilitating clearer comparisons across methods. Here, we compared training loss, and train/test accuracies across various learning rates for both CNN and QGCN. We train all models three times on different splits of the datasets.

Table 8 provides results collated across various train-test splits, in effort to capture model performance in different bias-variance regimes.

# G  Learning Rate Charts

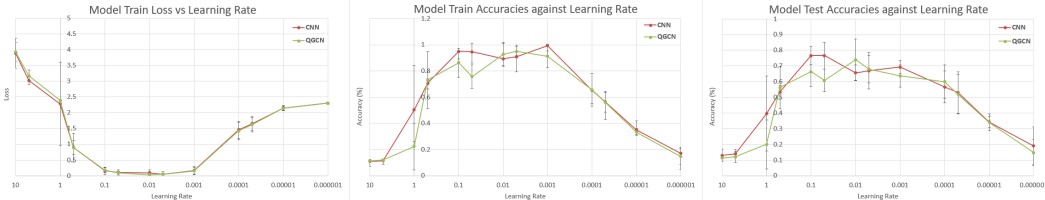

Figure 6: MNIST dataset results Figure for 100:20 dataset split

Table 8: *Standard Datasets*. CNN and QGCN model accuracies (mean ± S.D.) on standard datasets

| Dataset | Train:Test | Test Accuracy | |
| --- | --- | --- | --- |
| | | CNN | QGCN |
| MNIST | 100:20 | $76.67 \pm 5.77$ | $74.33 \pm 6.56$ |
| | 1000:200 | $95.50 \pm 0.71$ | $97.50 \pm 0.71$ |
| | 10000:1000 | $97.91 \pm 1.02$ | $97.57 \pm 0.69$ |
| | 60000:10000 | $98.92 \pm 0.10$ | $98.98 \pm 0.04$ |
| Fashion- | 100:20 | $72.67 \pm 2.31$ | $72.67 \pm 2.31$ |
| MNIST | 1000:200 | $84.67 \pm 0.03$ | $83.67 \pm 0.11$ |
| | 10000:1000 | $90.13 \pm 0.03$ | $90.23 \pm 0.26$ |
| | 60000:10000 | $92.56 \pm 0.18$ | $92.39 \pm 0.13$ |
| CIFAR-10 | 100:20 | $23.67 \pm 2.52$ | $24.33 \pm 2.31$ |
| | 1000:200 | $49.33 \pm 0.08$ | $47.67 \pm 0.20$ |
| | 10000:1000 | $68.08 \pm 0.45$ | $66.95 \pm 1.07$ |
| | 50000:10000 | $80.21 \pm 0.29$ | $79.59 \pm 0.35$ |

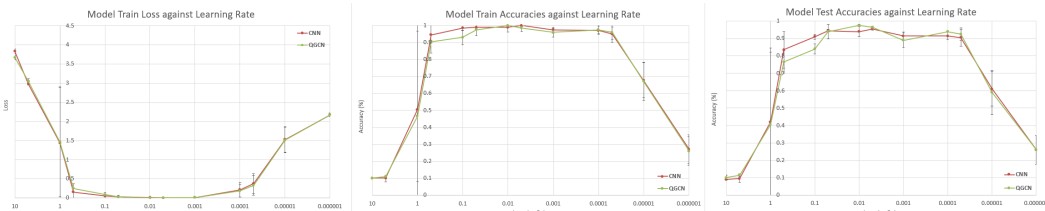

Figure 7: MNIST dataset results Figure for 1000:200 dataset split

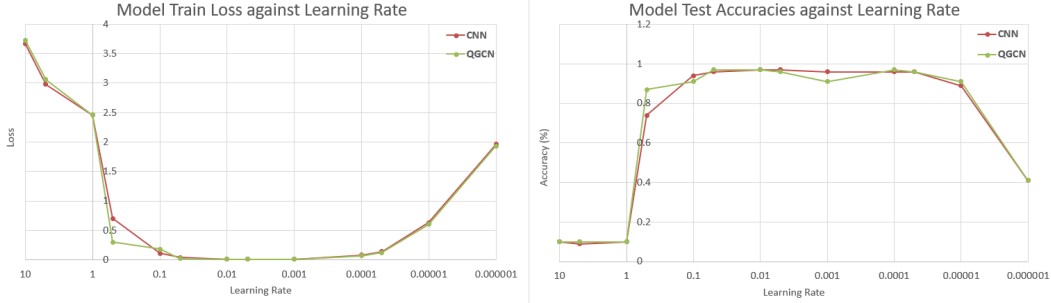

Figure 8: MNIST dataset results Figure for 10000:1000 dataset split

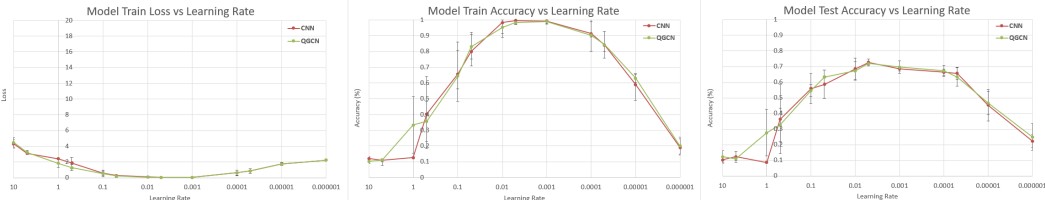

Figure 9: Fashion-MNIST dataset results Figure for 100:20 dataset split

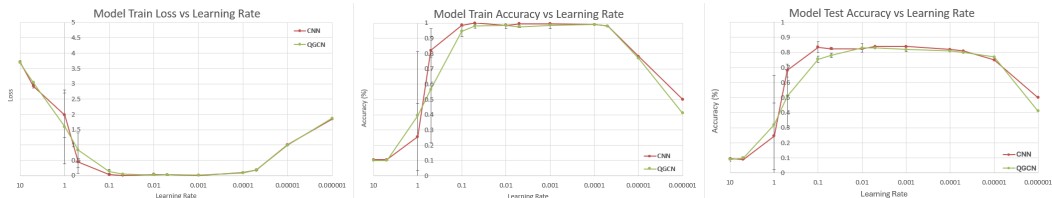

Figure 10: Fashion-MNIST dataset results Figure for 1000:200 dataset split

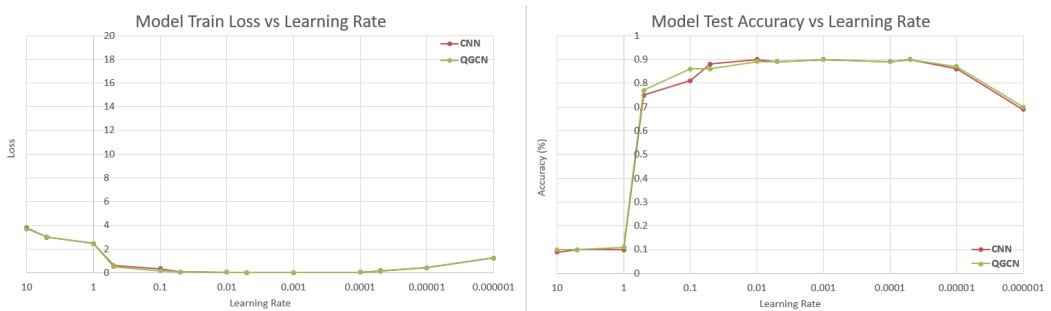

Figure 11: Fashion-MNIST dataset results Figure for 10000:1000 dataset split

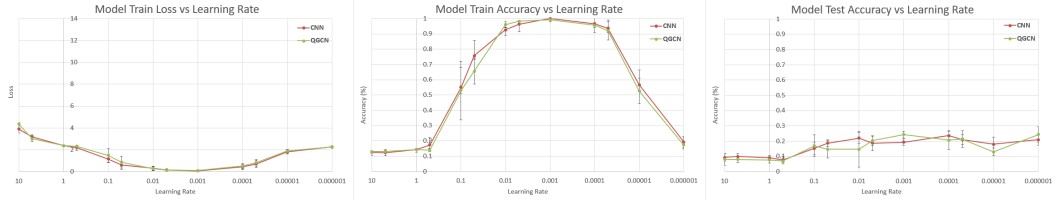

Figure 12: CIFAR-10 dataset results Figure for 100:20 dataset split

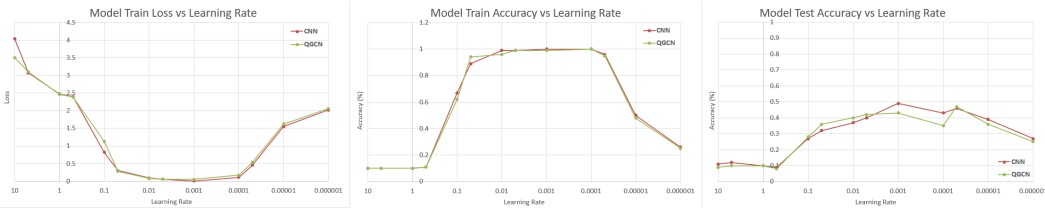

Figure 13: CIFAR-10 dataset results Figure for 1000:200 dataset split

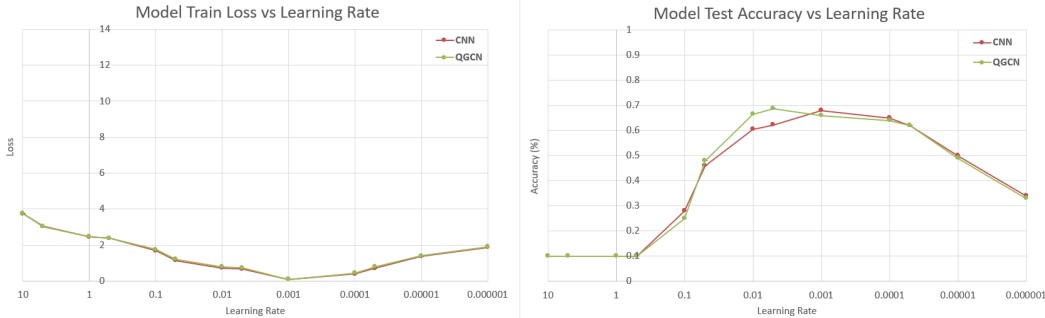

Figure 14: CIFAR-10 dataset results Figure for 10000:1000 dataset split

## H  Custom FEM Dataset

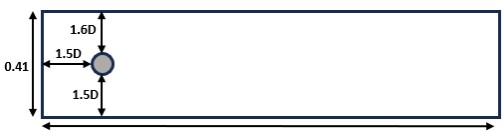

**(a)** Underlying 2D geometry and boundary conditions

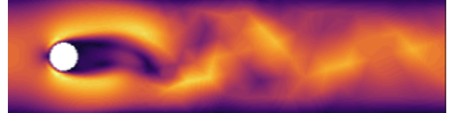

**(b)** Color map visualization of vortex shedding behavior for Re=120 (on an adaptive mesh with 1446 nodes and 2897 elements)

Figure 15: Underlying geometry and the flow around a circular cylinder.

We introduce several simulated Navier-Stokes non-linear dynamics benchmark datasets for both binary and denary classification tasks on FEM graphs. We simulated the "flow past a cylinder" problem on an adaptive mesh with the underlying two-dimensional flow geometry depicted in Figure 15a. We assumed a fluid density of $\rho = 1.0$ and dynamics governed by the time-dependent Navier-Stokes Equation:

$$u_t - v\Delta u + u\Delta u + \nabla p = 0, \nabla \cdot u = 0. \tag{11}$$

Here, $u$ represents velocity and $p$ represents pressure. We adopted a kinematic velocity of $v = 0.001$. For the lower and upper walls, as well as the boundary of the cylinder, no-slip boundary conditions were imposed. On the left edge, we prescribed a parabolic inflow profile as $u(0, y) = \left(\frac{4Uy(0.41-y)}{0.41^2}, 0\right)$ with a maximum velocity $U = \frac{3vRe}{2D}$. Here, $Re$ and $D$ denote the Reynolds number and the diameter of the cylinder, respectively. On the right edge, do-nothing boundary conditions define the outflow $v\frac{\delta u}{\delta n} - pn = 0$ with $n$ denoting the outer normal vector. We employed FEniCS library for solving the governing Navier-Stokes equations, using the adaptively refined mesh as shown in Figure 1b. for the spatial discretization in the finite element implementation. We conducted simulations to produce a dataset of flow velocities, covering a range of Reynolds numbers spanning values 20 to 120. At lower Reynolds numbers, the flow exhibits a stationary behavior. However, as the Reynolds number increases, a fascinating phenomenon known as Karman vortex shedding emerges. This phenomenon results in the flow adopting a time-periodic behavior, characterized by vortex shedding occurring behind the cylinder ([29, 34]). It's important to note that in our simulations, the primary flow direction is horizontal, emphasizing the significance of the x-component of velocity.

We generated the graph dataset representing fluid velocity components in a standard PyTorch Geometric format. We explored two inductive learning tasks with the graph dataset: binary and denary classifications. For binary, we separated laminar and turbulent flows based on distinct $Re$ values, while for denary classification, we used evenly spaced $Re$ values across a range. We created three datasets, as illustrated in Table 9: Navier-Stokes-Binary (NS-Binary) for binary classification and Navier-Stokes-Denary-1 (NS-Denary-1) and Navier-Stokes-Denary-2 (NS-Denary-2) for denary classification, with NS-Denary-2 being more challenging. Training data for each $Re$ value is from the

initial time period, and test data is from later time steps in the simulation, so models predict classes for future time points they have not seen.

Table 9: The table captures the different $Re$ ranges we considered for the different custom datasets and the step sizes. Notice in the binary case that $Re = 20\text{-}40$ are grouped in laminar class and 100-120 into the turbulent flow class.

| Dataset | Re range | Re step-size |
|---|---|---|
| NS-Binary | 20-40, 100-120 | 5 |
| NS-Denary-1 | 30-120 | 10 |
| NS-Denary-2 | 20-40, 100-120 | 5 |

Table 10: Shown in the table are different custom dataset splits we have provided as part of this paper. The third column captures the training time period per $Re$ from which train data were aggregated from the FEM time series solutions

| Dataset | Train:Test | Train Time (s) |
|---|---|---|
| NS-Binary | 100:20 | 0 - 0.005 |
| NS-Binary | 1000:200 | 0 - 0.05 |
| NS-Binary | 10000:1000 | 0 - 0.5 |
| NS-Binary | 20000:5000 | 0 - 1.0 |
| NS-Denary-1 | 100:20 | 0 - 0.005 |
| NS-Denary-1 | 1000:200 | 0 - 0.05 |
| NS-Denary-1 | 10000:1000 | 0 - 0.5 |
| NS-Denary-1 | 20000:2000 | 0 - 1.0 |
| NS-Denary-2 | 100:20 | 0 - 0.005 |
| NS-Denary-2 | 1000:200 | 0 - 0.05 |
| NS-Denary-2 | 10000:1000 | 0 - 0.5 |
| NS-Denary-2 | 20000:2000 | 0 - 1.0 |

# I  Benchmark Datasets for Graph Kernels

The diverse set of Benchmark Data Sets for Graph Kernels we used for validating QGRN and other GNN methods are as introduced below:

***AIDS***. 2000 sample binary classification dataset of molecular compounds with classes active and inactive, representing activity against HIV.

***COIL-DEL***. 3900 sample graph dataset extracted from images of 100 different objects taken at different poses (with pose interval of 5 degrees)

***Enzymes***. 600 sample size enzymes dataset with 6 classes (each of which represents one of the 6 EC top-level classes) from the BRENDA enzyme database.

***Letter (low/med/high)***. 2250 sample datasets of graphs representing distorted letter drawing with 15 classes, each corresponding to a different Roman alphabet letter. *low*, *med* and *high* represent different degrees of distortions applied to the hand-drawn letter graphs.

***Mutagenicity***. 4337 sample binary classification dataset of chemical compounds with classes mutagen and non-mutagen.

***Proteins*** and ***Proteins-Full***. 1113 sample size binary classification datasets for classifying protein graphs into enzymes or non-enzymes.

*Proteins-Full* differs in that it has many more node features: *Proteins*=3 and *Proteins-Full*=31

***Frankenstein***. A 4337 sample binary mutagenicity classification dataset modified from the BURSI dataset with classes, mutagen and non-mutagen. This dataset has continuous node features with very high dimensionality: 781 features per node.

***Mutag***. 188 sample binary classification dataset of chemical compounds and their mutagenic effect on bacterium

***Synthie***. A 400 sample Erdös-Rényi synthetic graphs dataset with 4 classes generated with probabilistic models. Figure 16 shows the distribution of the dataset samples across their different classes.

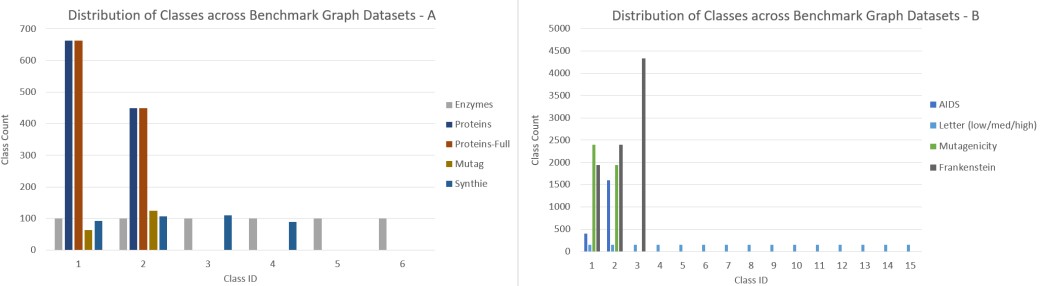

Figure 16: Distribution of samples by classes across all datasets used in this paper. We exclude COIL-DEL because it has 100 classes each with 39 samples and hence would render the plot illegible.

# J  Graph kernels benchmark datasets: additional results

Tables 11 and 12 contain additional results worth highlighting.

Table 11: *Standard datasets*. CNN and QGCN model accuracies on standard datasets

| Dataset | Train:Test | Train Accuracy | | Test Accuracy | |
| --- | --- | --- | --- | --- | --- |
| | | CNN | QGCN | CNN | QGCN |
| MNIST | 100:20 | $95.33 \pm 0.05$ | $92.67 \pm 2.31$ | $76.67 \pm 5.77$ | $74.33 \pm 6.56$ |
| | 1000:200 | $99.67 \pm 0.02$ | $99.67 \pm 0.02$ | $95.50 \pm 0.71$ | $97.50 \pm 0.71$ |
| | 10000:1000 | – | – | $97.91 \pm 1.02$ | $97.57 \pm 0.69$ |
| | 60000:10000 | – | – | $98.92 \pm 0.10$ | $98.98 \pm 0.04$ |
| Fashion-MNIST | 100:20 | $99.67 \pm 0.58$ | $98.33 \pm 1.15$ | $72.67 \pm 2.31$ | $72.67 \pm 2.31$ |
| | 1000:200 | $99.50 \pm 0.71$ | $97.05 \pm 0.71$ | $84.67 \pm 0.03$ | $83.67 \pm 0.11$ |
| | 10000:1000 | – | – | $90.13 \pm 0.03$ | $90.23 \pm 0.26$ |
| | 60000:10000 | – | – | $92.56 \pm 0.18$ | $92.39 \pm 0.13$ |
| CIFAR-10 | 100:20 | $96.67 \pm 4.93$ | $99.33 \pm 0.58$ | $23.67 \pm 2.52$ | $24.33 \pm 2.31$ |
| | 1000:200 | $99.33 \pm 0.33$ | $95.33 \pm 0.71$ | $49.33 \pm 0.08$ | $47.67 \pm 0.20$ |
| | 10000:1000 | – | – | $68.08 \pm 0.45$ | $66.95 \pm 1.07$ |
| | 50000:10000 | – | – | $80.21 \pm 0.29$ | $79.59 \pm 0.35$ |

Table 12: *Graph kernels benchmark datasets - III*. Test Accuracy (%) across different GCNs

| Models | Mutagenicity | Proteins-Full | Letters (low) | Letters (med) |
| --- | --- | --- | --- | --- |
| QGRN | $\mathbf{83.80} \pm 00.33$ | $\mathbf{87.13} \pm 00.55$ | $\mathbf{99.81} \pm 00.11$ | $\mathbf{96.76} \pm 00.63$ |
| GCNConv | $60.60 \pm 00.19$ | $85.15 \pm 00.86$ | $64.95 \pm 00.22$ | $41.52 \pm 01.39$ |
| ChebConv | $68.66 \pm 00.17$ | $85.15 \pm 00.99$ | $73.27 \pm 00.22$ | $63.18 \pm 00.40$ |
| GraphConv | $68.66 \pm 00.17$ | $85.48 \pm 00.57$ | $74.29 \pm 00.19$ | $64.25 \pm 00.90$ |
| SGConv | $60.60 \pm 00.11$ | $84.16 \pm 01.98$ | $65.21 \pm 00.11$ | $42.10 \pm 00.57$ |
| GENConv | $81.10 \pm 00.34$ | $85.48 \pm 00.57$ | $99.11 \pm 00.11$ | $90.92 \pm 00.67$ |
| GeneralConv | $68.81 \pm 00.11$ | $85.48 \pm 00.57$ | $74.29 \pm 00.19$ | $64.19 \pm 00.50$ |
| GATv2Conv | $80.02 \pm 00.33$ | $84.16 \pm 01.72$ | $96.70 \pm 00.55$ | $85.84 \pm 00.98$ |
| TransformerConv | $78.36 \pm 00.34$ | $85.15 \pm 00.72$ | $98.09 \pm 00.11$ | $90.10 \pm 00.50$ |

# K   Graph kernels benchmark datasets: model parameters and wall clocks

Table 13: *Graph kernels benchmark datasets - I*. Model sizes (number of parameters)

| Models | AIDS | Frankenstein | Mutag | Mutagenicity | Proteins | Proteins-Full |
| --- | --- | --- | --- | --- | --- | --- |
| QGRN | 64.86 | 1409.33 | 57.15 | 63.69 | 63.97 | 73.61 |
| GCNConv | 63.36 | 1437.89 | 57.54 | 62.59 | 62.79 | 68.16 |
| ChebConv | 64.13 | 1400.48 | 56.45 | 65.67 | 65.95 | 74.02 |
| GraphConv | 67.71 | 1447.30 | 56.45 | 66.69 | 66.95 | 74.11 |
| SGConv | 63.36 | 1408.83 | 57.54 | 62.59 | 62.79 | 72.61 |
| GENConv | 71.62 | 1399.17 | 60.80 | 60.80 | 60.99 | 74.82 |
| GeneralConv | 60.10 | 1393.67 | 59.33 | 59.33 | 59.52 | 77.51 |
| GATv2Conv | 64.13 | 1391.62 | 63.36 | 63.36 | 63.55 | 68.93 |
| TranformerConv | 64.90 | 1307.91 | 63.36 | 63.36 | 63.75 | 74.50 |

Table 14: *Graph kernels benchmark datasets - I*. Model sizes (number of parameters)

| **Models** | Synthie | Letters (h) | Letters (l) | Letters (m) | Enzymes | Coil-Del |
|---|---|---|---|---|---|---|
| QGRN | 68.19 | 61.51 | 58.21 | 58.21 | 69.71 | 70.34 |
| GCNConv | 65.54 | 60.21 | 60.21 | 60.21 | 66.57 | 61.09 |
| ChebConv | 69.96 | 59.15 | 59.15 | 59.15 | 71.81 | 75.56 |
| GraphConv | 70.53 | 59.15 | 59.15 | 59.15 | 71.81 | 75.56 |
| SGConv | 70.72 | 60.21 | 60.21 | 60.21 | 71.88 | 75.43 |
| GENConv | 73.28 | 62.67 | 62.67 | 62.67 | 74.18 | 73.64 |
| GeneralConv | 67.52 | 62.86 | 62.86 | 62.86 | 68.93 | 65.83 |
| GATv2Conv | 66.18 | 52.18 | 52.18 | 52.18 | 67.08 | 69.92 |
| TranformerConv | 68.87 | 53.88 | 52.88 | 52.88 | 70.50 | 70.12 |

Table 15: *Graph kernels benchmark datasets - I*. Google TPU Inference latency. Wall clock (in ms)

| **Models** | AIDS | Frankenstein | Mutag | Proteins |
|---|---|---|---|---|
| QGRN | $12.38 \pm 0.38$ | $17.50 \pm 1.02$ | $14.74 \pm 0.42$ | $15.69 \pm 2.28$ |
| GCNConv | $2.39 \pm 0.04$ | $3.04 \pm 0.32$ | $3.82 \pm 0.63$ | $2.42 \pm 0.14$ |
| ChebConv | $5.65 \pm 0.24$ | $5.53 \pm 0.37$ | $3.61 \pm 0.20$ | $6.25 \pm 0.43$ |
| GraphConv | $1.25 \pm 0.05$ | $1.81 \pm 0.19$ | $1.16 \pm 0.04$ | $1.94 \pm 0.60$ |
| SGConv | $4.20 \pm 0.09$ | $5.13 \pm 0.83$ | $4.10 \pm 0.26$ | $4.37 \pm 0.44$ |
| GENConv | $1.58 \pm 0.04$ | $1.89 \pm 0.13$ | $1.49 \pm 0.14$ | $1.57 \pm 0.16$ |
| GeneralConv | $1.67 \pm 0.06$ | $0.08 \pm 0.27$ | $1.66 \pm 0.31$ | $1.54 \pm 0.14$ |
| GATv2Conv | $3.33 \pm 0.27$ | $4.11 \pm 0.11$ | $3.34 \pm 0.21$ | $3.56 \pm 0.33$ |
| TransformerConv | $3.50 \pm 0.24$ | $4.06 \pm 0.34$ | $3.27 \pm 0.13$ | $4.09 \pm 0.44$ |

Table 16: *Graph kernels benchmark datasets - II*. Google TPU Inference latency. Wall clock (in ms)

| **Models** | Synthie | Letters (high) | Enzymes | Coil-Del |
|---|---|---|---|---|
| QGRN | $13.46 \pm 1.65$ | $10.83 \pm 0.20$ | $13.46 \pm 2.34$ | $13.68 \pm 0.99$ |
| GCNConv | $2.45 \pm 0.14$ | $2.26 \pm 0.06$ | $2.33 \pm 0.06$ | $2.37 \pm 2.37$ |
| ChebConv | $6.12 \pm 1.80$ | $3.88 \pm 0.28$ | $4.99 \pm 1.33$ | $3.65 \pm 0.08$ |
| GraphConv | $1.22 \pm 0.05$ | $1.25 \pm 0.06$ | $1.47 \pm 0.42$ | $1.26 \pm 0.14$ |
| SGConv | $4.05 \pm 0.19$ | $4.27 \pm 0.13$ | $4.32 \pm 0.23$ | $4.70 \pm 0.57$ |
| GENConv | $1.71 \pm 0.18$ | $1.54 \pm 0.07$ | $1.62 \pm 0.07$ | $1.51 \pm 0.09$ |
| GeneralConv | $1.47 \pm 0.06$ | $1.95 \pm 0.09$ | $1.54 \pm 0.12$ | $1.45 \pm 0.08$ |
| GATv2Conv | $3.37 \pm 0.31$ | $4.29 \pm 0.12$ | $3.29 \pm 0.15$ | $3.31 \pm 0.08$ |
| TransformerConv | $3.50 \pm 0.27$ | $4.63 \pm 0.19$ | $3.56 \pm 0.23$ | $3.78 \pm 0.17$ |

Table 17: *Graph kernels benchmark datasets - III*. Google TPU Inference latency. Wall clock (in ms)

| **Models** | Mutagenicity | Proteins-Full | Letters (low) | Letters (med) |
|---|---|---|---|---|
| QGRN | $24.88 \pm 9.04$ | $15.73 \pm 1.07$ | $14.12 \pm 2.53$ | $10.45 \pm 0.53$ |
| GCNConv | $3.07 \pm 0.32$ | $2.48 \pm 0.11$ | $3.50 \pm 0.75$ | $2.40 \pm 0.13$ |
| ChebConv | $4.81 \pm 0.21$ | $6.26 \pm 0.56$ | $3.70 \pm 0.16$ | $3.73 \pm 0.26$ |
| GraphConv | $1.27 \pm 0.02$ | $1.62 \pm 0.04$ | $2.10 \pm 0.75$ | $1.22 \pm 0.12$ |
| SGConv | $4.84 \pm 1.13$ | $5.57 \pm 1.22$ | $4.12 \pm 0.29$ | $4.03 \pm 0.11$ |
| GENConv | $1.50 \pm 0.03$ | $1.62 \pm 0.12$ | $1.42 \pm 0.05$ | $1.55 \pm 0.15$ |
| GeneralConv | $1.58 \pm 0.08$ | $1.56 \pm 0.18$ | $1.42 \pm 0.05$ | $1.55 \pm 0.15$ |
| GATv2Conv | $4.06 \pm 0.97$ | $3.35 \pm 0.10$ | $3.18 \pm 0.12$ | $3.30 \pm 0.11$ |
| TransformerConv | $4.12 \pm 1.09$ | $3.89 \pm 0.16$ | $3.34 \pm 0.27$ | $3.56 \pm 0.19$ |

## L    Graph Datasets: Node Classification

Tables 18 and 19, representing homophilic and heterophilic dataset results respectively, provide the full set of datasets, against which we benchmarked, for node classification tasks.

Table 18: *Homophilic node classification datasets*. Test Accuracy (%) across different GCNs

| Models | Photo | Computers | Cora | PubMed | CiteSeer |
|---|---|---|---|---|---|
| QGRN | $95.34 \pm 0.10$ | $90.02 \pm 0.02$ | $\mathbf{89.02} \pm 0.14$ | $\mathbf{89.11} \pm 0.15$ | $\mathbf{79.09} \pm 0.27$ |
| GraphConv | $94.44 \pm 0.04$ | $87.96 \pm 0.16$ | $87.15 \pm 0.44$ | $88.39 \pm 0.07$ | $76.69 \pm 0.07$ |
| GENConv | $95.25 \pm 0.04$ | $\mathbf{91.66} \pm 0.05$ | $86.31 \pm 0.36$ | $87.73 \pm 0.19$ | $75.37 \pm 0.34$ |
| GeneralConv | $94.13 \pm 0.14$ | $89.29 \pm 0.02$ | $87.64 \pm 0.04$ | $88.97 \pm 0.09$ | $75.53 \pm 0.10$ |
| EGConv | $\mathbf{96.19} \pm 0.05$ | $91.50 \pm 0.06$ | $88.34 \pm 0.30$ | $88.38 \pm 0.08$ | $76.34 \pm 0.21$ |

Table 19: *Heterophilic node classification datasets*. Test Accuracy (%) across different GCNs

| Models | Chameleon | Squirrel |
|---|---|---|
| QGRN | $74.15 \pm 0.37$ | $56.17 \pm 0.45$ |
| GraphConv | $72.77 \pm 0.32$ | $64.25 \pm 0.09$ |
| GENConv | $71.56 \pm 0.67$ | $58.00 \pm 0.18$ |
| GeneralConv | $\mathbf{78.11} \pm 0.29$ | $\mathbf{66.80} \pm 0.08$ |
| EGConv | $63.54 \pm 0.07$ | $48.44 \pm 0.41$ |

## M    Leader Board - Comparison with QGRNs

Our search for state of the art performance was limited to Papers with Code, which has only a subset of the datasets we trained on. This, we believe, is indicative of the fact that in literature, a subset of these benchmarks are chosen for different types of downstream tasks. Our method mostly focused on inductive classification tasks. After searching thoroughly through Papers with Code, we found what we believe to be the full subset of data sets with comparable SOTA results below:

Table 20: *Leader-board (Papers with Code)*. Comparison of QGRN performance to leading models

| Dataset | Average Test Accuracy (%) | | Leading Model Name |
|---|---|---|---|
| | QGRN | Leading Model | |
| AIDS | **99.50** | 97.30 | k-NN classifier: IAM Repository |
| MUTAG | **100.00** | **100.00** | Evolution of Graph Classifiers |
| Mutagenicity | **83.80** | 83.00 | Tree-G |
| Proteins | 80.20 | **84.91** | HGP-SL |
| Enzymes | 72.50 | **78.39** | DSGCN-allfeat-2020 |
| Frankenstein | 75.58 | **78.90** | GWL_WL (Graph Invariant Kernels) |

## N    IAM Graph Database - Comparison with QGRNs

The conference paper, "IAM Graph Database Repository for Graph Based Pattern Recognition and Machine Learning" ([30]), provides k-NN classifier-based results that the authors intended as "a first reference system to compare other algorithms with". Clearly these do not represent SOTA results, but we include them as another baseline comparison available for the data sets considered in our paper:

Table 21: *IAM Graph Database Repository*. Comparison of QGRN performance to k-NN classifier

| Dataset | Average Test Accuracy (%) | |
|---|---|---|
| | QGRN | k-NN |
| Letters (low) | **99.81** | 99.60 |
| Letters (medium) | **96.76** | 94.00 |
| Letters (high) | **94.10** | 90.00 |
| Coil-Del | **94.14** | 93.30 |
| AIDS | **99.50** | 97.30 |
| Mutagenicity | **83.80** | 71.50 |
| Proteins | **80.20** | 65.50 |

# O  Training Deeper QGRNs

We present some preliminary results on training deeper QGRNs, in an effort to understand the model's ability to overcome GCNs inability to go deep. In this exploration, we trained different depths of the same QGRN model each on 2 different datasets. These datasets roughly capture the extremes of dataset difficulty in the paper. We trained on the AIDS dataset: a small binary classification 1600:400 train:test split dataset, and the Letters (high) dataset: a more complex dataset with 15 classes and train:test = 1725:525 split. We trained these models with ADAM optimizer and cross entropy loss, under the same conditions as used elsewhere in the paper (see Sec. 4.3).

Table 22: *Deeper QGCN and QGRN networks*. Sample results illustrating impact of deeper network on model performance

| Dataset | Model Depth | Model Size (K) | | Mean Test Accuracy (%) | |
|---|---|---|---|---|---|
| | | QGCN | QGRN | QGCN | QGRN |
| AIDS | 3 | 30.91 | 59.43 | 99.25 | **99.50** |
| | 6 | 59.62 | 101.26 | **99.25** | **99.25** |
| | 9 | 88.32 | 143.09 | **99.25** | **99.25** |
| | 12 | 117.03 | 184.92 | **99.25** | 99.00 |
| | 18 | 174.43 | 268.58 | **99.25** | 99.00 |
| Letters (high) | 3 | 30.90 | 56.33 | 92.95 | **94.10** |
| | 6 | 59.60 | 95.07 | 91.24 | **93.33** |
| | 9 | 88.30 | 133.81 | 89.57 | **91.19** |
| | 12 | 117.01 | 172.55 | 90.23 | **91.24** |
| | 18 | 174.42 | 250.05 | 87.24 | **90.86** |

**Discussion**: From the results compiled in table 22, we do see that as the depth of the network increases there is a modest reduction in overall model performance. The results also reveal that the extent of model performance regression depends on the complexity of the dataset. The AIDS dataset is less sensitive to model depth: scaling the model depth by 6x (from 3 to 18) results in just a 0.5% loss in mean test accuracy. Letters (high), on the other hand, sees more performance loss at the same 6x depth, i.e., 5% loss in performance. Though part of the regression could be attributed to insufficient number of epochs for the deeper networks, but we speculate that it is largely due to our residual network retrofit in our QGRN layer not being sufficient to mitigate the effects of depth (e.g., the vanishing gradient problem). There are several exciting findings in the literature that have suggested methods such as customized aggregators instead of standard addition and softmax aggregation in the message passing network (layer level innovations). Others approach this problem architecturally, e.g., borrowing from successful architectures such as ResNets (as we did), using skip connections, pooling and sampling layers, drop-out inspired methods, and so on. We highlight that our paper's focus is primarily extending CNN's convolution operation to arbitrary graphs, which we demonstrate is the case through our equivalence analysis. In this context, we highlight that deep vanilla CNNs also face this same challenge of being difficult to train. In the world of CNNs, this problem is combated via architectural innovations such as batch normalization, use of better activations like ReLU, skip connections etc. In this light, our layer-level innovation with quantizable neighborhoods (replicating

CNN-like convolution) also stands to benefit from such architectural designs that allow for robust and deeper network architectures, which we leave to future work.

## P  Performance impact of quantization

Here, we explore a pilot comparison of satisficing (angular) mapping and QuantNet to understand how quantizing the neighborhoods uniformly vs learning the quantization affects model performance. We were able to compile together some sample results that provide some insight into this, which we share in Table 23. In Table 24, we considered the impact of using pseudo-positional descriptors (here, the entire node attributes) as opposed to explicit spatial descriptors as positional descriptors.

Table 23: *Deeper satisficing mapping (SM) and QuantNet networks.* Sample results illustrating impact of deeper network on model performance

| Dataset | Model Depth | Model Size (K) | | Mean Test Accuracy (%) | |
|---|---|---|---|---|---|
| | | SM | QuantNet | SM | QuantNet |
| AIDS | 3 | 30.91 | 59.43 | 99.25 | **99.50** |
| | 6 | 59.62 | 101.26 | **99.25** | **99.25** |
| | 9 | 88.32 | 143.09 | **99.25** | **99.25** |
| | 12 | 117.03 | 184.92 | **99.25** | 99.00 |
| | 18 | 174.43 | 268.58 | **99.25** | 99.00 |
| Letters (high) | 3 | 30.90 | 56.33 | 92.95 | **94.10** |
| | 6 | 59.60 | 95.07 | 91.24 | **93.33** |
| | 9 | 88.30 | 133.81 | 89.57 | **91.19** |
| | 12 | 117.01 | 172.55 | 90.23 | **91.24** |
| | 18 | 174.42 | 250.05 | 87.24 | **90.86** |

Table 24: *3-layer QGRN model analysis.* Sample results for QGRN model trained with and without positional descriptors.

| Dataset | Average Test Accuracy | |
|---|---|---|
| | With Positional Descriptors | Without Positional Descriptors |
| AIDS | **99.50** | 99.50 |
| Coil-Del | **94.57** | 94.14 |
| Letters (high) | **94.10** | **94.10** |
| Letters (med) | **97.14** | 96.76 |
| Letters (low) | **100.00** | 99.81 |

**Discussion**: In Table 23, we examined 2 different datasets representing the extremes of classification difficulty: AIDS (a binary classification task with 2000 samples) and Letters-high (a dataset with 15 classes and 2250 total samples). We notice from Table 5 that in the easier inductive task, both uniform and learned quantization methods yield similar performance. Learned quantization outperforms uniform quantization in the harder learning task, Letters (high), where the ability to learn which bins activation input elements should be clustered, becomes beneficial. We also observe that the relative model performance improvement of 1-2% persists between the satisficing (angular) mapping and QuantNet, even with increasing model depth, adding more validation to quantization learning as a useful property. In Table 24, we focused only on QuantNet-based QGRN and trained the model on a sampling of datasets which had positional attributes. We prepared from the same datasets, variants with positional attributes and variants without positional attributes for the quantization learning. Intuitively, we'd expect that with quantization learning from explicit positional descriptors, binning should be close to optimal, thereby leading to optimal model performance. This is what the experimental results show. Without explicit positional descriptors (here QGRN uses the entire node attributes, of which explicit positional attributes are inclusive), the model achieves near-optimal test accuracies. It is interesting to point out that the model performance of the variants without positional descriptors is upper bounded by the model performance on variants with positional descriptors.

# Q Empirically determining quantization bins

The number of bins is set by choosing the number of subkernels, which is user defined and hence a hyper-parameter of our model. As such, the number of bins can be tuned via any hyper-parameter optimization method, such as Grid search, Bayesian search etc. For image datasets, this choice is immediately obvious: the number of bins must be the total number of neighboring nodes in any given neighborhood (8 neighbors + 1 central node). For the uniform quantization case, we developed an algorithm (an algorithm pseudo-code, Algorithm D.1) to determine satisficing mapping across all neighborhoods. For QGRN with learnable quantization, the number of bins is a hyper-parameter. We present a sample hyper-parameter search for select graph datasets from the TUDatasets benchmark in Table 25.

Table 25: *Number of bins - hyper-parameter search*. Hyper-parameter search of optimal number of bins/subkernels for QGRN

| Dataset | Average Test Accuracy (%) | | | | |
|---|---|---|---|---|---|
| | Bins=2 | Bins=3 | Bins=5 | Bins=7 | Bins=9 |
| AIDS | 99.25 | 99.25 | 98.75 | **99.50** | 99.25 |
| Enzymes | 70.83 | 71.67 | 69.17 | **72.50** | 68.89 |
| Coil-Del | 91.26 | 92.39 | 92.39 | **94.14** | 90.54 |
| Letters (high) | 93.91 | 93.71 | 93.71 | **94.67** | 93.71 |
| Proteins | 73.27 | 76.24 | 73.27 | **80.20** | 73.27 |

**Discussion**: Notice the classification test accuracy variations as we sweep the number of bins/subkernels. The trend is non-linear, mostly peaking between 3 - 7 bins/subkernels. This is partly explained by the fact that most datasets here have an average neighborhood size of 5 - 7, which provides a reasonable partitioning size for all neighborhoods; the analogy here is CNN's bin size of 9, which is a consequence of full regular neighborhoods having a size of 9 pixels. As the number of subkernels grow, each subkernel sees a smaller subset of the training data; additionally, bin data size distribution imbalance results in uneven training of the corresponding bin subkernels, which results in regression of the overall model's performance. A pathological case worth mentioning here would be defining an arbitrarily large number of bins (far greater than the dataset's average neighborhood size), leading no or very little training of subkernels: the weights that see any node features are bound to overfit on the small set of node features they see while those not seeing any node features will never get to train. During test time, this blend of over-fitted and under-fitted subkernel weights will result in large regression in the overall model's performance. On the other extreme, with a single learnable subkernel, all types of nodes which ideally may belong to different bins are collapsed into the same bin, making it hard for the single subkernel to learn common messages across the entire node set of the graph for message passing. This is an example of underfitting (high bias) scenario. The optimal bin size is intuitively expected to be around the average neighborhood size of the given dataset, which is what we see in the experimental results summarized in Table 25.

# R Compute resources for experiments

All experimental results provided in this paper were the results of runs on Google Colab premium offering of GPUs and TPUs. As much as specific details on these systems are publicly available, we have tabulated them below. The two main accelerator options we used were the A100 GPUs and TPUs. In table 26, we outline some of the system specifications. It is worth noting that the set of datasets we explored in the TUDataset benchmark were all moderately sized, hence Google Colab's Pro+ offering was more than sufficient for us. This was also the case for results collated on the standard dataset splits and our own custom Navier-Stokes FEW dataset. Not included in the table is Tesla V100, which at the time of finalizing this paper, Colab had deprecated support for it.

# S Addressing Latency Concerns

Regarding model training and inference speed, there is much room for future work to develop new algorithms for choosing subkernel masks that are faster than those we introduce here. We see the

Table 26: *Compute Resources*. Google Colab Pro+ offering

| System | System RAM (GB) | GPU RAM (GB) | Disk Space (GB) |
|---|---|---|---|
| Nvidia A100 | 83.5 | 40.0 | 201.2 |
| Google T4 GPU | 51.0 | 15.0 | 201.2 |
| Google L4 GPU | 62.8 | 22.5 | 201.2 |
| Google TPU | 334.6 | NA | 225.3 |

primary contribution of this paper as the introduction of the quantized convolutions theoretical framework, which we demonstrated with 2 options for subkernel selection:

1. 2D angular-quantization (satisficing mapping).
2. Flexible learnable-quantization (QuantNet).

There are many possible algorithms for choosing sub-kernel masks within this framework, and we suspect that future research into this area could be very fruitful. For example, some practical ways satisficing mapping & QuantNet might be sped up include:

1. Separating out tensor operations in the message preparation, propagation and update stages of the message-passing neural networks (MPNN) and leveraging the operator fusion capability of Torch JIT Script to optimize these operation sets.
2. Parallelizing the execution of the sub-kernel convolutions, with dedicated low level CUDA kernels, instead of using grouped convolutions (as we do in our current implementation).
3. Using depth-wise separable convolutions: this will reduce the model complexity (in terms of number of parameters, hence resulting in a proportional reduction in model runtime complexity. Depth-wise separable convolutions trade off model flexibility for model size. This means this optimization would need to be carried out carefully to ensure that QGCN/QGRN doesn't regress significantly in performance. It is worth noting that CNNs also have been sped up in this way for edge platforms.

The above listed are by no means exhaustive but we believe these will be good starting points for optimizing QGCN/QGRN to make them competitive with existing highly optimized MPNNs in the literature. We do hope that further research into QGCN/QGRNs will allow them to eventually be useful for numerous applications (as happened historically for CNNs over time).

# T  Impact statement

As the paper's primary innovation is a learning model that can be trained to perform different tasks on graphs, it by itself doesn't pose any obvious risks. The potential for negative societal impact depends on the dataset and downstream learning tasks a user of this model designs for. We strongly recommend for such a model to be used in compliance with all ethical standards appropriate to the domain in which it is targeted to be deployed. We accentuate the need to ensure that datasets on which this model is trained reflect not only the status quo but take into account broader social, political, economical and religious contexts so that they mitigate potential harms from any potential bias or misuse of the model's learning abilities.

