# OpenReview forum: "Generalizing CNNs to graphs with learnable neighborhood quantization"
_NeurIPS.cc/2024/Conference — NeurIPS 2024 poster_

### Official Review · Reviewer_XG75 · 2024-07-02

**Soundness:** 3
**Presentation:** 3
**Contribution:** 3
**Rating:** 6
**Confidence:** 4

**Summary:**

In this paper, the authors introduce a Quantized Graph Convolution Network (QGCNs), which directly extends CNN on GCN by decomposing the convolution operation into non-overlapping sub-kernels. It shows that a QGCN is identical to a 2D CNN layer on a local neighborhood of pixels. Then, they generalize this approach to graphs of arbitrary dimension by approaching sub-kernel assignment as a learnable multinomial assignment problem. Meanwhile, they also integrate the network with a residual operation and achieve better performance than the current GCN models.

**Strengths:**

Compared with other CNN-based models extending on graph datasets, the authors first quantizing the convolution kernel into an equivalent set of non-overlapping sub-kernels applied to graph geometry. In my opinion, the novelty of this manuscript is relatively novel. Besides, the presentation of this paper sounds relatively clear and logical.

**Weaknesses:**

1) The motivation seems unclear. As shown in the abstract and introduction, the authors attempt to extend CNN on graph datasets. However, in my view, GCNs have achieved excellent performance on graph tasks. Why do the authors integrate the convolution kernel into the GCN network?
2) The downstream task of the proposed network is ambiguous. Does it solve the image classification or graph-related tasks? The authors conduct them both in the experiment section.
3) The manuscript lacks some reference to the baseline model in the experiment and some notations lack definitions.
4) The experiment settings are not detailed enough to reproduce the experiment results.

**Questions:**

1) In my opinion, GCN has achieved excellent performance on graph datasets. Why do you integrate the convolution kernel into the GCN network to handle the graph data?
2) Why conduct the experiments on image datasets like MNIST, Fashion-MNIST or CIFAR-10? Does it verify that QGCN and 2D CNN have the same performance? In my view, it should be conducted on graph scenarios.
3) I suggest that the author could describe the downstream task of the proposed model in detail.
4) In line 150, the authors state that they want the output to be a graph as well. However, for most GCN-based models, the output is always the representation of the node or edge. Hence, why does the proposed graph layer output a graph?
5) In Section 3.2, I am curious about how to handle the scenario that more than 2 neighbors fall in the same bound.
6) As shown in Line 199, the complexity of the proposed satisficing mapping is $O(|V|^2)$. Is there a strategy to reduce the complexity?
7) The authors should describe more experiment settings like learning rate, network layers, and so on.
8) The manuscript lacks some reference to the baseline model in the experiment and some notations lack definitions.

**Limitations:**

It is an interesting idea that extend CNN into GNN and handle the graph datasets. The authors could describe the motivation in more detail and clearly, which makes the manuscript more convincing. Meanwhile, although the authors first quantizing the convolution kernel into an equivalent set of non-overlapping sub-kernels applied to graph geometry, the complexity is too high such as $O(|V|^2)$. Besides, The authors should adjust the experiment section and add more descriptions about the experiment setting.

---

> ### Author Rebuttal · Authors · 2024-08-06
>
> Responses to listed weaknesses (*W*) and question (*Q*):
>
> *W1* and *Q1*. We agree that we have done an inadequate job at explaining our motivation for this work in the introduction. We will update the introduction and related work sections to clarify our motivation and why we desired to improve on existing models. In brief, our original motivation was to create a model that did a better job taking advantage of positional information in graphs by generalizing CNNs (for example, doing better than SGCNs) by handling positional information in a more flexible way. In pursuing this line of research, we further discovered that our method also substantially outperformed GCNs on the graph benchmark datasets that we report in the manuscript.
>
> *Q2*. We agree that we should have been more clear about why we are sometimes considering image data when our main practical application target is graph data. In brief, there are two main threads in the results section: **a.** empirical validation of theory: the results reported in table 1 were meant to demonstrate that a 2D-CNN on image data and a QGCN using satisficing mapping on graphs generated by connecting local pixels result in equivalent classification performance. These results are meant to empirically back up our theoretical results that show QGCNs reduce to 2D-CNNs on image data (i.e., to demonstrate that QGCNs properly generalized CNNs by reducing to CNNs on image data); **b.** performance on graph data: the remaining tables show QGCN and our extensions of it are highly performant against SGCN on spatial graph data (Table 2; also including graph based representations of image data as benchmarks as in prior work) and against other state-of-the-art GNNs and graph data benchmarks (Tables 3 and 4). We think that (a) is interesting theoretically and (b) is of practical interest for graph data. We will make this distinction much more clear in our revision of the manuscript.
>
> *W2* and *Q3*. Thank you for the feedback, we will update section 4 and the appendices in detail to more clearly describe the downstream tasks for each of our experiments. We will highlight that all tasks currently described in section 4 are framed as graph classification tasks when using QGCN, where images are transformed into graphs by connecting adjacent pixels. CNNs are applied directly to image data. Again, the image tasks are included to address point **a.** above.
>
> *W3*. We will carefully clarify in detail references to baseline models in our experiments ensure all notations have clear definitions.
>
> *W4*. We will add code to reproduce all experiments including our Navier-Stokes dataset exactly with seeds set for exact reproducibility.
>
> *Q4*. The reviewer is correct that the output of Eqn. (3) is a node representation. We will clarify our language on line 150 to point out that after each layer, the node representations are embedded in a graph with the same edge structure as the input graph.
>
> *Q5*. In section 3.2, the satisficing mapping is actually defined such that no 2 neighbors will ever fall within the same bound (see lines 190-191).
>
> *Q6*. Yes! In the case when all neighborhoods have the same positional structure, the satisficing mapping only needs to be calculated once and can then be cached, resulting in a substantial speedup (comparable to CNN).
>
> *Q7. / Q8.* Thank you for pointing this out. We will update the manuscript to clarify experimental settings, references, and notation. If you have any suggestions for specific edits we could make, we will be happy to incorporate them!
>
> We hope this response was thorough enough to address the concerns you have raised. Thank you again for your time to review this manuscript.

---

### Official Review · Reviewer_1wWD · 2024-07-07

**Soundness:** 3
**Presentation:** 4
**Contribution:** 3
**Rating:** 7
**Confidence:** 3

**Summary:**

The authors find a new a way to generalize CNN from Euclidean space to graph data. The proposed method has good analogy with CNN's computing pattern, and can be applied to data with/without positional descriptors.
On dataset with positional descriptors, the proposed method is equivalent to SGCN. On dataset without positional descriptors, the proposed method matchs or outperforms all GNN methods across adiverse graph learning problems.

**Strengths:**

All designs of the proposed method are derived from authors' insights into CNNs and are thus well motivated. The proposed method has a better analogy with CNNs than existing GNNs in terms of spatial computing pattern, and also has a stronger expressive power.

Experiments are comprehensive and convincing. The proposed method is compared with a wide range of baselines on multiple datasets.

The manuscript is well-written and easy to follow. I can go through the paper without any confusion.

**Weaknesses:**

My major concern is how is the proposed method's performance compared with the state-of-the-art results of each dataset. Are the sota baselines included in table 3 and table 4? Table 17 shows some sota baseline's results, but only covers a limited number of datasets.

A minor concern is that forward/backward speed of the proposed method may be slow, but this has been discussed by the authors.

**Questions:**

1. Why the leading accuracy of AIDS dataset in Table 17 is lower than the baselines' accuracy in table 3?

2. Please add a bracket to eq. (2) to clarify the order of operations.
$$
O_{c,d,:} = \sum_{h=0}^{JK-1} \color{red}\left[\color{black} \left(W\sum\sum ...\right) + B \color{red}\right]\color{black}
$$

**Limitations:**

the authors have adequately addressed the limitations.

---

> ### Author Rebuttal · Authors · 2024-08-06
>
> Responses to listed weaknesses (*W*) and question (*Q*):
>
> *W1.* Tables 3 and 4 in the paper record the performance of QGRN as compared to similar sized models in the GNN literature. A singular simple architecture subsuming these layers were trained across different hyperparameter ranges, as documented in the paper and codebase, and the best performance on the benchmark was reported for each model. In regards to Table 17, our search for state of the art performance on this benchmark was limited to Papers with Code, which has only a subset of the datasets we trained on. This, we believe, is indicative of the fact that in literature, a subset of these benchmarks are chosen for different types of downstream tasks. Our method mostly focused on inductive classification tasks. After searching thoroughly through Papers with Code, we found what we believe to be the full subset of data sets with comparable SOTA results below:
>
> | Dataset | QGRN Avg. Test Acc. (%) | SOTA Avg. Test Acc. (%) | Leading Model Name |
> |-----------|-----------|-----------|-----------|
> | Aids |  **99.50**  |  97.30  | k-NN classifier: IAM Graph Database Repository (2008)   |
> | Mutag |  **100.00** | **100.00** | Evolution of Graph Classifiers |
> |  Mutagenicity |  **83.80** | 83.00 | Tree-G |
> |  Proteins | 80.20 | **84.91** | HGP-SL |
> | Enzymes | 72.50 | **78.39** | DSGCN-allfeat-2020 |
> |  Frankenstein | 75.58 | **78.90** | GWL_WL (Graph Invariant Kernels) |
>
> In addition, the conference paper “IAM Graph Database Repository for Graph Based Pattern Recognition and Machine Learning”, published in 2008 (also cited in our paper) provides some k-NN classifier-based results that the authors intended as “a first reference system to compare other algorithms with”. Clearly these **do not** represent SOTA results, but we include them as another baseline comparison available for the data sets considered in the table below:
>
> | Dataset | QGRN Avg. Test Acc. (%) | kNN Avg. Test Acc. (%) |
> |-----------|-----------|-----------|
> | Letters (low) | **99.81**| 99.60 |
> | Letters (medium) | **96.76**| 94.00 |
> | Letters (high) | **94.10** | 90.00 |
> | Coil-Del | **94.14** | 93.30 |
> | AIDS | **99.50** | 97.30 |
> | Mutagenicity |  **83.80** |  71.50 |
> | Proteins |  **80.20** | 65.50 |
>
> *W2.* Thank you for pointing this out. We agree and add further response on this topic in the general rebuttals section above.
>
> *Q1.* Thank you for highlighting this. We noticed this was odd, especially given that the k-NN classifier approach from 2008 was beating a novel approach from 2017. After examining DGCNN’s manuscript, we found that though the reported number is classification accuracy, it was for a different dataset (coincidentally bearing the same name as TUDatasets AIDs on Papers with Code). DGCNN’s AIDs dataset is a 3-class classification dataset, while TUDatasets' AIDs is a binary classification dataset. Thank you for helping us catch this. On Papers with Code, we found that there was a placeholder leaderboard for the AIDS dataset we trained on (named “AIDS Antiviral Screen”), with no benchmarks or papers submitted for it. In the absence of any submissions, we have reported the k-NN classifier accuracy as the best to date that we know of, but would consider removing the row from Table 17 at the reviewer's request.
>
> *Q2.* Concern is duly noted. We have identified some clarity concerns and plan to thoroughly clean notations up in order to improve readability and digestibility of the paper’s contents.
>
> We hope we have adequately addressed the reviewer's concerns and thank the reviewer for their time.

---

> > ### Comment · Reviewer_1wWD · 2024-08-14
> > **Thanks for you reply. I will keep my score.**
> >
> > It seems that there is still a gap between your method and SOTA, but considering the clear motivation behind, I will keep my score.

---

### Official Review · Reviewer_rsM6 · 2024-07-13

**Soundness:** 2
**Presentation:** 3
**Contribution:** 2
**Rating:** 6
**Confidence:** 1

**Summary:**

In this paper, the authors present a novel Quantized Graph Convolution Layer (QGCL) that extends the benefits of CNNs’ strong local inductive bias to graphs. The authors have shown that embedding a QGCL within a residual network architecture give state-of-the-art results on benchmark graph datasets.

**Strengths:**

1. The presentation of the paper is clear and easy to follow.
2. Extending the benefits of CNNs’strong local inductive bias to graphs is interesting and instructive, which has the potential to advance the understanding and analysis of Graph Neural Networks.
3. The method is effective in comparison to baseline methods.

**Weaknesses:**

1. Implementation of QGCN is not yet efficient as demonstrated in the comparison of inference time with other baselines.

**Questions:**

1. Is the efficiency of QGCN related to the average degree of nodes in the graph?
2. How does QGCN’s training time compare to baseline methods?

---

> ### Author Rebuttal · Authors · 2024-08-06
>
> Responses to listed weaknesses (*W*) and question (*Q*):
>
> *W1.* Thank you for highlighting this. We kindly refer to the related response given in the general rebuttal section, as this was a repeated concern for most reviewers. Thank you.
>
> *Q1.* Runtime efficiency is indeed determined partly by the average node degree of the input graphs. The primary factor that determines the runtime efficiency is the number of sub-kernels initialized in a QGRL for its convolution. This is because our current implementation of QGRL convolution effectively serializes the individual sub-kernel convolutions, thereby forcing the runtime to roughly scale linearly with the number of subkernels used. Some secondary factors determining runtime efficiency would be the average degree of nodes in the graph, input graph node feature size, the dimension of positional descriptors, which will all collectively contribute to the model's size. On model efficiency, a variety of factors work together to influence this. The paper already carries out different sensitivity analysis (outlined in-depth in the appendix) on, e.g., how the number of subkernels influence model performance, how the type of quantization (here, QGCN vs QGRN) impacts model performance etc. We will expand and highlight these results more prominently in the revision.
>
> *Q2.* Currently, QGCN’s training time is comparable to CNN, where the local neighborhoods are regular, i.e., positional topologies are the same across all graphs. This is already highlighted in the paper (and we plan to emphasize it better in our revision). Our approach here caches the quantized sub-kernel masks, i.e., the satisficing mapping of nodes to sub-kernels, once it is computed so that subsequent epochs do not have to recompute the same mapping. We note that caching, as a runtime optimization strategy, isn’t novel to our method; some popular GNNs use this approach, e.g., GCN, which caches its renormalization matrix. For QGRN, our current implementation is not optimal due to the serialization of sub-kernel convolutions (note this is not required but rather a product of our code not yet being fully optimized). As such, QGRN runs slower by roughly a factor of *k* (where *k* = number of sub-kernels used in the network’s convolution) compared to other models (particularly with respect to GCN or GAT, for example). This is also mentioned in the paper, but we shall emphasize it more so that it is apparent.
>
> We hope that this response (along with the general response above) was informative and addressed your concern about efficiency. We thank you for your time.

---

> > ### Comment · Reviewer_rsM6 · 2024-08-12
> > **I thank authors for comments. Will keep my score.**
> >
> > Thank you for taking the time to address my concerns. After reading the other reviews and answers I maintain my review score.

---

### Official Review · Reviewer_5Ti4 · 2024-07-13

**Soundness:** 2
**Presentation:** 2
**Contribution:** 2
**Rating:** 5
**Confidence:** 3

**Summary:**

In this paper, the authors propose Quantized Graph Convolution Networks (QGCN), a GCN framework that directly extends CNNs by decomposing convolution operations into non-overlapping sub-kernels. This paper demonstrate that QGCN is essentially the same as a 2D CNN layer in dealing with pixel local neighbourhoods, and generalize the approach to graphs of arbitrary dimensions. After integrating the algorithm into a residual network architecture, the algorithm demonstrates performance that matches or outperforms other state-of-the-art GCNs on several benchmark datasets.

**Strengths:**

1. This article completely extends CNNs to GNNs, and it is an important task to build better architectures for processing graph data.
2. The QGCN algorithm proposed in the article achieves quite excellent results on Benchmark datasets for graph kernels.
3. This paper proposes a simple but effective method that computes the difference between the target and source features, and then project that difference onto a vector representing the assignment weights of each subkernel. This approach is somewhat enlightening.

**Weaknesses:**

1. The authors experiment only on Benchmark datasets for graph kernels. Node classification and edge classification tasks are also very important graph tasks, but there is no mention of related experiments in this paper.
2. The baselines that the authors compare are a little old-fashioned, such as GAT, which was proposed in 2017, and there is a lack of comparison with the most novel approaches.
3. The heavy computational burden of GNN hinders its practical application, however, the Inference latency of QGRN is about 6 times higher than that of GCN, which raises concerns about the application of QGRN in the real world.

**Questions:**

1. For hyperparameter $\phi$, can the authors give the results of its sensitivity analysis?

**Limitations:**

The authors adequately addressed the limitations of thier work.

---

> ### Author Rebuttal · Authors · 2024-08-06
>
> Responses to listed weaknesses (*W*) and question (*Q*):
>
> *W1*. Excellent point. To address other important graph tasks, in addition to adding the SVAE example above we now run several node classification tasks on multiple types of datasets, mostly citation networks (like Cora, PubMed), Wikipedia hyperlinks networks (like such as the Chameleon dataset) and product relations networks (such as with the Amazon Photos, Computers etc.). We highlight that the datasets used in this exploration exhibit different degrees of homophily and heterophily properties. Chameleon and Squirrel datasets exhibit strong heterophily while all the others exhibit stronger homophily. Given the brief time to investigate this, we have only pursued **limited architectural exploration** and here chose a single architecture that proved reasonably performant across all the models we compared against. The final architecture had the structure below:
>
> **Conv**>**BatchNorm**>**Relu** > **sum**( 3x(**Conv**>**Relu**) ).
>
> The convolution (also referred to here as ‘conv’) layers are where the different graph convolutional layers like QGRL, GenConv, EGConv etc. plug into. This network has 2 layers, the first is a convolution layer followed by batch normalization and then a ReLU activation layer. The second layer in the architecture sums up features from three identical blocks, each of which is a convolution layer followed by a ReLU activation layer. We trained all models across a range of learning rates (0.1, 0.05, 0.01, 0.005, 0.001) for about 2000 epochs and mimicked early stopping by caching the model state that produced the largest validation set accuracy. Given the now apparent fact that MPNNs (Message Passing Neural Networks) generally struggle with heterophily datasets, we designed the generic architecture with edge directionality awareness, as inspired by the paper “Edge Directionality improves Learning on Heterophilic Graphs” by Rossi et. al. Empirically, we noticed that most of the performance improvement came from training on the reversed edge index tensor, which is equivalent to training on the original graph dataset, except with reversed edge directions. This is intuitive as a commonality across these datasets is the fact that a small number of nodes have high degrees and hence reversing edge directionality allows nodes with fewer edges to learn better from the representations of the few popular nodes. We note that QGRN performs competitively on the homophilic datasets. On the heterophilic datasets, except for Squirrel (the most heterophilic, where we suspect further fine tuning can help us bridge the performance gap), QGRN performs moderately well. Thus, beyond the strong results presented in our main paper, our method also shows potential for node/edge classification in both homophilic and heterophilic graphs.
>
> Table 1. Node Classification: Comparing QGRN to other recent models.
> | | Photo | Computers | Cora | PubMed | CiteSeer | Chameleon |  Squirrel |
> |-----------|-----------|-----------|-----------|-----------|-----------|-----------|-----------|
> | QGRN | 95.34 ± 0.10 | 90.36 ± 0.02 | **89.02 ± 0.14** | **89.11 ± 0.15** | **79.09 ± 0.27** | 74.15 ± 0.37 | 56.17 ± 0.45 |
>  | GraphConv |  94.44 ± 0.04 |  87.96 ± 0.16 | 87.15 ± 0.44 | 88.39 ± 0.07 | 76.69 ± 0.07 | 72.77 ± 0.32 | 64.25 ± 0.09 |
> | GenConv |  95.25 ± 0.04 |   **91.66 ± 0.05** | 86.31 ± 0.36 | 87.73 ± 0.19 | 75.37 ± 0.34 | 71.56 ± 0.67 |  58.00 ± 0.18 |
> | GeneralConv |  94.13 ± 0.14 | 89.29 ± 0.02 | 87.64 ± 0.04 | 88.97 ± 0.09 | 75.53 ± 0.10 | **78.11 ± 0.29** | **66.80 ± 0.08** |
> | EGConv|  **96.19 ± 0.05** | 91.50 ± 0.06 | 88.34 ± 0.30 |  88.38 ± 0.08 | 76.34 ± 0.21 |  63.54 ± 0.07 | 48.44 ± 0.41 |
>
>
> *W2.* Thank you for highlighting this. We did consider a variety of models in our work. Please note that for GAT, we actually compared against the improved GAT model, released in 2021, not the 2017 version. Additionally, we considered other transformer-based models like TransformerConv. Newer models like GNNDLD, reported to be beating SOTA on many node classification tasks (on Papers with Code), don't have publicly available codebases yet, hence the absence of any comparison with them. Please find the table below outlining the various models we compared against and the year these models were publicly released:
>
> | *Model* | GCNConv | ChebConv | GraphConv | SGCN | SGConv | GenConv | GeneralConv | TransformerConv | GATv2Conv | EGConv |
> |-----------|-----------|-----------|-----------|-----------|-----------|-----------|-----------|-----------|-----------|-----------|
> | *Year* |  2016 | 2016 | 2018 | 2019 | 2019 | 2020 | 2020 | 2020 | 2021 | 2022
>
>
> *W3.* Please see our response in the general rebuttal section.
>
> *Q1.* We fully expect that the impact of 𝜙 will be negligible in the vast majority of datasets, and primarily include it in the description of the model to make for easier visual interpretation of which nodes get assigned to which sub kernels when a large portion of nodes occur at an angle of 0 (such as in the case of graphs constructed from local neighborhoods in 2D image data). There will be a few cases in which the choice of 𝜙 might influence performance, such as when the majority of edges in a dataset are clustered around a specific angle. However, we do not readily have access to any such datasets to provide a meaningful sensitivity analysis of 𝜙. A more sensitive related parameter is the number of subkernels *k*. The paper already carries out a sensitivity analysis (outlined in-depth in the appendix) on, e.g., how the number of subkernels influence model performance, how the type of quantization (here, QGCN vs QGRN) impacts model performance etc. We will highlight these sensitivity analyses more explicitly in our revision of the paper as we agree such analyses are important.
>
> We hope we have adequately addressed the reviewer's concerns and that they will consider increasing their score accordingly. Thank you for your time.

---

> ### Comment · Reviewer_5Ti4 · 2024-08-08
>
> Thanks to the authors for your patient reply. I recognize the potential of GRCN in the field where throughput time is not significant. Besides, the experimental results presented in Table 1 show that GRCN is very effective in node classification, especially for datasets such as Cora, Citeseer, Pubmed. However, I have to point out that the accuracy of GraphConv & GenConv on the Cora, Citeseer and Pubmed datasets is higher than my usual experience.
> In general, the author's reply eliminated part of my concern. I am open to improve my score.

---

> > ### Author Response · Authors · 2024-08-08
> >
> > Thank you for engaging with our rebuttal.
> >
> > On your concern with *'the accuracy of GraphConv & GenConv on the Cora, Citeseer and Pubmed datasets is higher than my usual experience'*,  we'd like to affirm that it was indeed similar observations we made when we trained various GNNs like GraphConv and GenConv, without edge-directionality awareness. Dir-GNN (published in 2023) highlights the benefits of edge-directionality awareness on model performance. Dir-GNN is not a GNN but instead a wrapper around GNNs, non-invasively introducing edge-directionality awareness into the models being wrapped. Once we replicated this in our architecture, we saw that GNNs like GraphConv, GenConv, EGConv etc., and our own model, began to see a drastic improvement in model performance. We hope this clarifies the reviewer's doubts.
> >
> > Thank you for being open to improving your scores.

---

> ### Comment · Reviewer_5Ti4 · 2024-08-12
>
> I would like to thank the authors for addressing my concern, no further questions from my side, I'll increase my score of to 5.

---

### Author Rebuttal · Authors · 2024-08-06

We would like to thank the reviewers for dedicating their time and providing high quality feedback on our manuscript.

One concern noted in various ways by the reviewers was that the computational overhead of QGCNs and QGRNs is high. We agree that this is a limitation of our approach as currently implemented. This primarily limits these models in applications that require very low latency and high throughput. Importantly, there are many applications in need of expressive graphical models such as ours where such speed concerns are not significant.

As one example, a primary motivation behind these models was the need for more expressive ways to model brain networks from EEG data in a way that takes spatial information into account. We plan to publish a clinically-focused manuscript that uses these models for psychiatric clinical trial research in the near future at a clinical-neuroscience-focused venue. We have put together a quick demonstration for the reviewers using the publicly available [DEAP dataset](http://www.eecs.qmul.ac.uk/mmv/datasets/deap/), where we compare the performance of a supervised autoencoder using either QGRN layers or SGCN layers to construct the encoder and decoder. Methods and results for that experiment are reported below and will be added to the revised manuscript’s supplement (with code for reproducibility). We note that the inference time of GNNs, our model inclusive, is inconsequential in a clinical setting, even for real-time feedback applications where a new window of EEG is processed every few seconds.

The FEM example in the manuscript is another application where throughput time is not significant, as the tens of milliseconds required for inference are virtually instantaneous compared to the hours to days required for a full FEM simulation with a physical model. We will edit the paper to more clearly point out the FEM application and other applications that can currently benefit from QGCNs. Training times may still be significant for some of these applications, but we are unaware of any other models that can achieve the desired level of performance on the types of datasets we are interested in.

Another point regarding model speed is that there is much room for future work to develop new algorithms for choosing subkernel masks that are faster than those we introduce here. We see the primary contribution of this paper as the introduction of the quantized convolutions theoretical framework, which we demonstrated with 2 options for subkernel selection:
- 2D angular-quantization (satisficing mapping, QGCN)
- flexible learnable-quantization (QuantNet, QGRN)

There are many possible algorithms for choosing sub-kernel masks within this framework, and we suspect that future research into this area could be very fruitful. For example, some practical ways satisficing mapping & QuantNet might be sped up include:
* Separating out tensor operations in the message preparation, propagation and update stages of the MPNN and leveraging the operator fusion capability of Torch JIT Script to optimize these operation sets.
* Parallelizing the execution of the sub-kernel convolutions, with dedicated low level cuda kernels, instead of using grouped convolutions (as we do in our current implementation).
* Using depth-wise separable convolutions: this will reduce the model complexity (in terms of number of parameters, hence resulting in a proportional reduction in model runtime complexity. Depth-wise separable convolutions trade off model flexibility for model size. This means this optimization would need to be carried out carefully to ensure that QGCN/QGRN doesn’t regress significantly in performance. It is worth noting that CNNs also have been sped up in this way for edge platforms.

The above listed are by no means exhaustive but we believe these will be good starting points for optimizing QGCN/QGRN to make them competitive with existing highly optimized MPNNs in the literature. We do hope that further research into QGCN/QGRNs will allow them to eventually be useful for numerous applications (as happened historically for CNNs over time).

Other concerns involved benchmarking our methods for node/edge classification and clarifying comparisons against SOTA results. We directly address these by including new results and tables in the responses to reviewers 5Ti4 and 1wWD below.

### EEG Supervised Autoencoder Example

The last 42 s of each recording in the DEAP EEG dataset were divided into sliding windows of 3 s with 50% overlap, then were *z*-scored and spectral power was calculated in four frequency bands for each of the 32 electrodes. Power features were used as node attributes in a fully-connected graph containing all 32 electrodes. We trained a separate model for each subject (64%/16%/20%: training/val/test splits). The generative objective was MSE and the supervised objective was cross-entropy loss for classifying whether subjects self-rated their emotional state as positive valence or negative valence. Models were pre-trained for 1000 epochs on just the generative objective, then for another 100 epochs with one of 3 values of weight on the classification objective [100, 1000, 10000]. Validation sets were used to select the weight and number of training epochs with highest area under the receiver operating curve (AUC).

We found that using QGRN layers in the same autoencoder architecture compared to SGCN layers resulted in better generative and supervised loss values on the held out test sets. In addition, the QGRN-based model resulted in significantly better classification performance for this difficult classification problem (as measured by AUC). The mean +/- SEM over all subject test sets are reported in the following table:

| | QGRN | SGCN |
|-----------|-----------|-----------|
| MSE loss (Generative) |**1787.33 ± 315.51** | 2169.38 ± 317.80 |
| Cross-entropy loss (Supervised) | **0.6500 ± 0.0088** | 0.6591 ± 0.0091 |
| AUC |  **0.593 ± 0.014**  |  0.562 ± 0.011 |

---

### Decision · Program_Chairs · 2024-09-25

**Decision:**

Accept (poster)

**Comment:**

All four reviewer expressed enthusiastic support to this work, and an accept is recommended.